# TAD border deletion at the *Kit* locus causes tissue-specific ectopic activation of a neighboring gene

Evelyn Kabirova [1,3], Anastasiya Ryzhkova[1,3], Varvara Lukyanchikova[1,3], Anna Khabarova[1,3], Alexey Korablev[1], Tatyana Shnaider[1], Miroslav Nuriddinov [1], Polina Belokopytova[1,2], Alexander Smirnov[1], Nikita V. Khotskin[1], Galina Kontsevaya[1], Irina Serova[1] & Nariman Battulin [1,2] ✉

Topologically associated domains (TADs) restrict promoter-enhancer interactions, thereby maintaining the spatiotemporal pattern of gene activity. However, rearrangements of the TADs boundaries do not always lead to significant changes in the activity pattern. Here, we investigated the consequences of the TAD boundaries deletion on the expression of developmentally important genes encoding tyrosine kinase receptors: *Kit, Kdr, Pdgfra*. We used genome editing in mice to delete the TADs boundaries at the *Kit* locus and characterized chromatin folding and gene expression in pure cultures of fibroblasts, mast cells, and melanocytes. We found that although *Kit* is highly active in both mast cells and melanocytes, deletion of the TAD boundary between the *Kit* and *Kdr* genes results in ectopic activation only in melanocytes. Thus, the epigenetic landscape, namely the mutual arrangement of enhancers and actively transcribing genes, is important for predicting the consequences of the TAD boundaries removal. We also found that mice without a TAD border between the *Kit* and *Kdr* genes have a phenotypic manifestation of the mutation − a lighter coloration. Thus, the data obtained shed light on the principles of interaction between the 3D chromatin organization and epigenetic marks in the regulation of gene activity.

Genome folding into topologically associated domains (TADs) and compartments is considered to be an important level of gene regulation, affecting the frequency of contacts between regulatory elements and gene promoters. TADs as functional units have been proposed to prevent unwanted promoter-enhancer contacts and narrow down the enhancer search space. The functional importance of TADs for gene regulation was emphasized by analyzing disease-associated mutations, disrupting their boundaries. Structural variants, crossing TAD boundaries, can provoke pathogenic gene misexpression through the rewiring of promoter-enhancer interactions (see review[1,2]). Deletions have the potential to cause TADs fusion, resulting in ectopic contacts.

In T cell acute lymphoblastic leukemia disruption of CTCF-associated boundary allows the proto-oncogene activation[3]. Another example is limb malformation, triggered by deletion of a TAD boundary at the *Ihh/Epha4/Pax3* locus[4]. Inversions and translocations were shown to cause TADs shuffling[4–7]. In some cases, this shuffling triggers an enhancer adoption followed by ectopic activation of the gene[4]. In other studies, the pathogenic effect of the inversion results from the loss of expression of one of the genes with no enhancer adoption[8]. Overall, the susceptibility of a gene to an aberrant activation by an ectopic enhancer is different and partially depends on the gene's chromatin status[6].

[1]Institute of Cytology and Genetics SB RAS, Novosibirsk, Russia. [2]Novosibirsk State University, Novosibirsk, Russia. [3]These authors contributed equally: Evelyn Kabirova, Anastasiya Ryzhkova, Varvara Lukyanchikova, Anna Khabarova. ✉e-mail: battulin@gmail.com

Although altered TADs architecture is indeed associated with some pathogenic phenotypes[9,10], genome wide disappearance of TADs insulation upon CTCF or cohesin depletion does not cause global transcriptional effects[11,12]. Besides, in a number of studies TADs fusion had only mild effects on gene expression pattern, as enhancers inside the newly formed TAD retained their specificity, for example in *Kcnj2−Sox9* locus[13] and in *Shh* locus[14]. Unlike the deletion of CTCF sites, the repositioning of a TAD boundary by an inversion redirected enhancer activity and induced gene misexpression, highlighting the role of an entire TAD substructure with its epigenetic landscape in altering gene regulation[13]. Overall, how the regulatory environment inside the TAD affects the output of a boundary disruption remains an open question.

To test the effect of TAD boundary loss in different regulatory contexts, we investigated the *Kit* locus. The region comprises developmental regulators *Pdgfra*, *Kit* and *Kdr*, each located in the corresponding TAD, demarcated by strong domain boundaries. Mutations at this locus are often associated with malignant processes. Amplification of *Pdgfra, Kit* or *Kdr* is common among glioblastomas[15,16]. In a subset of gliomas, *Pdgfra* is aberrantly activated by an ectopic enhancer, which is associated with reduced CTCF binding[10]. *Kit* and *Pdgfra* mutations, resulting in the constitutive activation of these genes, are often found in gastrointestinal tumors[17]. Aberrant activation of *Kit* and *Pdgfra* by the oncogene-induced super-enhancers is essential for leukemic cell growth and survival[18]. Finally, *Kdr* is considered one of the most critical pro-angiogenic factors in tumor angiogenesis. *Kdr* expression is upregulated in cancer cells through an autocrine loop with its ligand[19]. Thus, maintaining the integrity of the chromatin structure within the *Kit* locus appears to be important, since misinteraction of enhancers with proto-oncogenes can activate oncogenic signals.

Here, we generated a series of genome-edited mice with deletions of boundary and intra-TAD CTCF sites and used primary cell cultures to track the effect of boundary perturbations. Using cell types with contrasting activity of *Pdgfra*, *Kit* and *Kdr* allowed us to test the role of boundary insulation in different regulatory contexts. The level of architectural changes varied from a marginal increase of inter-TAD contacts across the Pdgfra/Kit boundary, to the disrupted insulation between Kit/Kdr TADs. Loss of insulation between Kit and Kdr TADs resulted in cell type-specific ectopic gene activation. The boundary disruption was associated with a specific phenotype in mice − a lighter coat color. Collectively, our findings support the idea that the structural and functional outcome of CTCF binding sites (CBSs) deletions depends on the epigenetic landscape of neighboring TADs.

## Results

### Regulation of the genes *Pdgfra, Kit, Kdr* is tissue-specific

The *Kit* gene has been attracting the attention of researchers for a long time, as its mutations induce a clearly visible dominant white-spotting phenotype[20–22]. *Kit* as well as its flanking genes *Pdgfra* and *Kdr* encode receptor tyrosine kinases, which participate in fundamental cellular processes in metazoans[23], such as signal transduction from growth factors and cytokines, cell proliferation and differentiation, etc. Consistent with their function, each of the genes features a distinct tissue-specific expression. *Pdgfra*-positive cells can be found in a wide array of mesenchymal tissues[24–26]. *Kit* is expressed among several tissues, particularly in mast cells and melanocytes. Our transcriptome analysis reveals *Kit* being among the most expressed genes in mast cells (Fig. 1a). *Kdr* is an endothelial-specific receptor and its expression is almost exclusively restricted to the blood vessels[27,28]. Our data (Fig. 1a) is consistent with the expected expression patterns.

The contrast expression of the genes makes the *Kit* locus a good model to evaluate an enhancer hijacking. We focused on three cell types: fibroblasts (*Pdgfra* is expressed, *Kit* is inactive), mast cells (*Kit* is among top 0.3% of the expressed genes, *Kdr* is inactive), and

melanocytes (*Kit* is among top 3% of the most expressed genes, *Kdr* is weakly expressed).

### Three adjacent TADs define 3D structure of the Kit locus

To characterize the TADs organization we performed capture Hi-C (cHi-C) in mouse embryonic fibroblasts (MEFs), mast cells and melanocytes. Our data revealed that the region is organized into three adjacent TADs with *Pdgfra*, *Kit* and *Kdr* each located in a separate TAD (Fig. 1b).

Notably, the *Pdgfra* gene body overlaps an extended region between Pdgfra and Kit TADs, while its promoter and the first exon are located immediately upstream of the Pdgfra TAD boundary. As to *Kit* and *Kdr*, their coding regions are located within the corresponding TADs.

The Kit TAD is heterogeneous in spatial contacts with two distinct self-interacting regions in mast cells and melanocytes. We noted that in mast cells the insulation between these regions appears to coincide with the *Kit* terminator, while in melanocytes it corresponds to the *Kit* promoter. The observed insulation could be associated with transcription, as it has been shown to interfere with cohesin loop-extrusion[29].

We asked whether the observed 3D genome organization of the locus − with three adjacent TADs, one for each of the genes − is conserved across vertebrates. It is worth mentioning that though a mechanism of using SMC proteins for the genome folding is conservative[30], as might be averaged 3D contacts in different cell types[31] or certain 3D patterns, such as the second diagonal on Hi-C maps of erythroid cells in vertebrates[32], but details of the mechanism may vary between species and the exact genome coordinates of TADs vary not only between species, but in single cells of an organism. Consequently, we introduce a limitation and by conservativity we understand that *Pdgfra*, *Kit*, and *Kdr* are (1) in close genomic proximity, (2) each located within a separate TAD.

To estimate the evolutionary conservativity of TADs in the *Kit* locus, we liftovered Hi-C contacts from several vertebrate species to the mouse genome, using C-InterSecure software[33], since a direct comparison of Hi-C maps from different species is inaccurate due to different genome sizes (Supplementary Fig. 1a). Resulting interaction maps demonstrate a high similarity of chromosome architecture across the *Kit* locus in human, mice, dog, chicken, rabbit[34], and african clawed frog[35] (Supplementary Fig. 1b). As we mentioned, the non-conservative nature of TADs at the precise genomic location varies among different species. However, the consistent spatial organization observed in the *Kit* locus implies its evolutionary importance, aligning with the critical role these genes play in early development.

The fact that *Pdgfra*, *Kit* and *Kdr* genes were established from a common ancestor through a series of duplications[23] explains why genes with similar function are located at the same locus. But their tissue-specific activity demands an independent regulation. H3K27ac data reveals how regulatory context varies between mast cells, melanocytes and MEFs (Fig. 1c). Here we refer to enhancers based on enrichment of an enhancer-related epigenetic mark H3K27ac. In mast cells enhancers are found within the Kit TAD upstream of the *Kit* gene. In melanocytes enhancers within the Kit TAD surround the *Kit* gene. In MEFs enhancers are only found upstream of the *Pdgfra* gene. The Kdr TAD lacks H3K27ac enrichment in all three cell types. We assumed that TADs are crucial for insulating regulation at the locus.

We acknowledge H3K27ac enrichment is a non-exhaustive definition of an enhancer. To additionally characterize the regulatory context we addressed data from other studies on chromatin accessibility, enhancer-associated (H3K4me1) and heterochromatin-associated (H3K27me3) epigenetic marks, as well as binding of transcription factors (Supplementary Fig. 2). H3K27ac-associated enhancers were confirmed to locate in open chromatin in all the cell types, and for mast cells it demonstrated that H3K27me3 spans all the

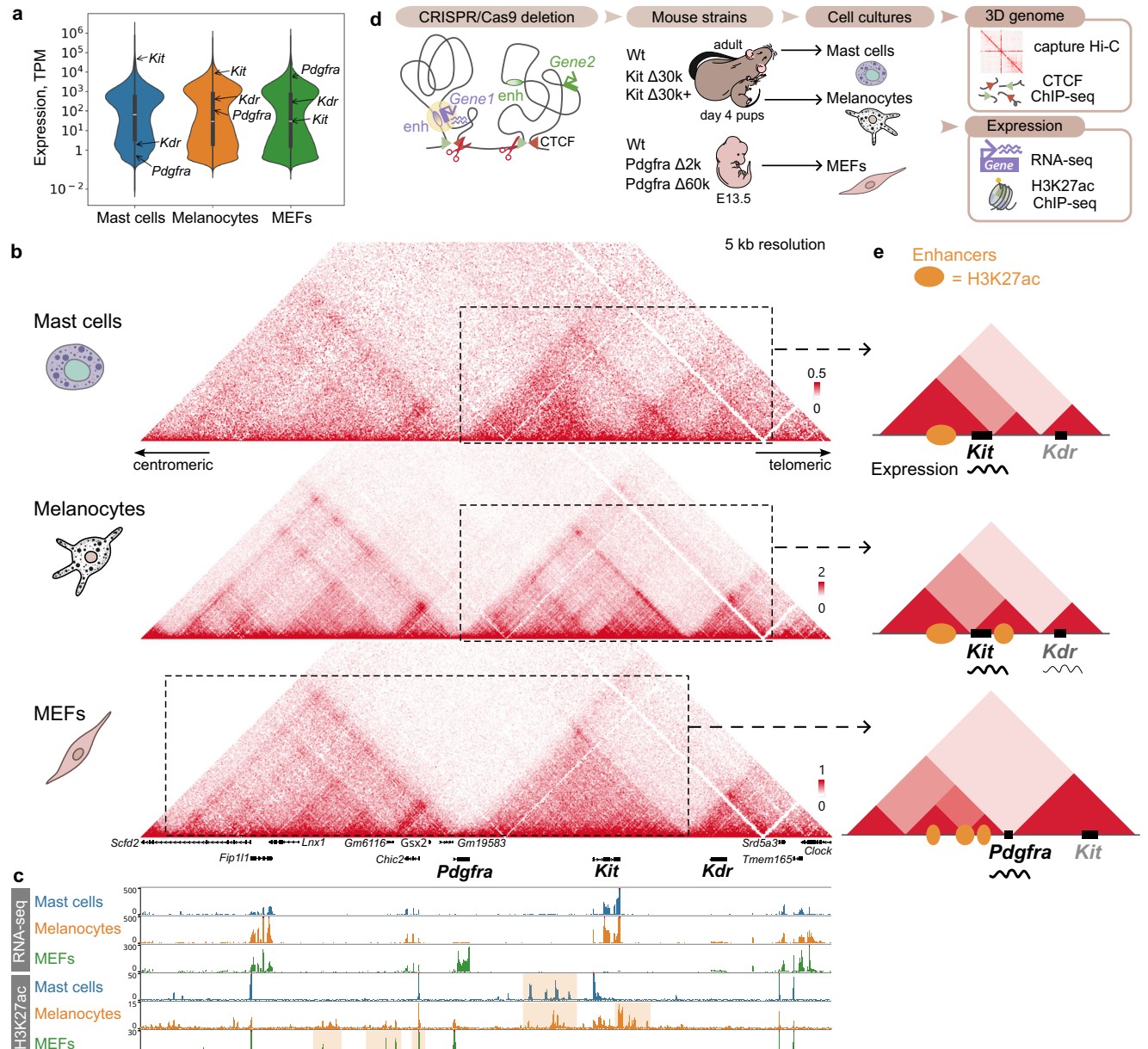

**Fig. 1 | Spatial and regulatory genome characteristics of the *Kit* locus. a** Gene expression in mast cells, melanocytes and mouse embryonic fibroblasts (MEFs) revealed by RNA-seq (*n* = 3 independent primary cell cultures per group). The height of a column: an expression level in transcripts per million (TPM), the width of a column: an amount of genes with the exact expression level. Genes *Pdgfra*, *Kit*, *Kdr* demonstrate contrasting expression in the chosen cell types. The center mark in the box represents the median, bounded by the 25th and 75th percentiles, and the whiskers are the most extreme data point, which is no more than 1.5 times the interquartile range. Source data are provided as a Source Data file; **b** Hi-C maps (5 kb resolution) demonstrate that genes *Pdgfra*, *Kit*, *Kdr* are each located in a distinct TAD with minor features being specific to a cell type. Color bars reflect the interaction counts; **c** Expression and regulatory context of the locus in the cell types. Regions which we refer to as enhancers are highlighted yellow in H3K27ac track; **d** Research design. Using CRISPR/Cas9 we generated mutant mouse strains with a TAD boundary deletion. The mice were used to obtain primary cell cultures for a further analysis of 3D genome organization and gene expression; **e** Schematic view of the tissue-specific TAD structures with emphasis on enhancer localisation relative to genes and expression level (silent genes are coloured gray).

locus downstream from *Kit* gene. We believe that by expanding regulatory characterisation of the locus by other studies we provide more certainty to the enhancers assignment, though with a limitation that we refer to them as an enhancer upstream or downstream of a gene rather than an exact enhancer-promoter interaction.

Importantly, enhancers of the *Kit* gene were functionally estimated in another study through a series of deletions[36]. Two enhancer regions were identified upstream of *Kit* in mast cells and melanocytes (located at 147–154 kb and 21–28 kb, respectively). While distant regions were required for *Kit* expression in mast cells (upon their

deletion *Kit* was not transcribed), a more proximal regulatory region was essential in melanocytes.

We expected the removal of the borders to result in TADs fusion, permitting enhancer hijacking. To assess whether TADs restrict an enhancer influence to its target promoter, we generated a series of mouse strains carrying different deletions of the key CBSs (Fig. 1d). We obtained primary cell cultures from the mice. Utilizing homogenous cell populations (Supplementary Fig. 3) provides an advantage of acquiring more precise functional genomic data compared to obtaining an averaged picture from animal tissues. Primary cell cultures were

used to analyze spatial genome organization using cHi-C and CTCF ChIP-seq, and gene expression using H3K27ac ChIP-seq and RNA-seq.

Comparing the effects of CBSs deletions in different cell types allowed us to analyze an impact of different regulatory contexts on the same locus (Fig. 1e schematically demonstrates the tissue-specific differences of the *Kit* locus).

### *Pdgfra* gene localisation at the TAD border challenges the border disruption

*Pdgfra* and *Kit* show a contrasting expression in MEFs, i.e. *Pdgfra* is actively transcribed, while *Kit* is not transcribed (Fig. 1a). According to our hypothesis, we expected TADs fusion in the absence of boundary CBSs to result in an abnormal activation of *Kit* by the *Pdgfra* enhancers.

The Pdgfra/Kit border area comprises borders of two adjacent TADs separated by a 50 kb region with the *Pdgfra* gene body located in that space. We characterized the architecture of the border based on our ChIP-seq data and further consensus motif identification. The Pdgfra border consists of three strong CBSs, one forward and two reverse-facing, that overlap the *Pdgfra* gene body (Fig. 2a, b). The Kit border consists of one forward-facing CBS.

To disrupt the Pdgfra/Kit border we generated two mouse strains: Pdgfra Δ2k, and Pdgfra Δ60k. Using CRISPR/Cas9 we deleted the CBSs at the Pdgfra border (Pdgfra Δ2k) and the inter-TAD region up to the CBS at the Kit border (Pdgfra Δ60k). Both deletions did not affect the promoter and the first exon of the *Pdgfra* gene. We compared the effect of the deletions on the spatial genome organization with wild-type (Wt) cHi-C maps (Fig. 3a).

The Pdgfra Δ2k mouse strain carries a 2 kb deletion of the three CTCF peaks at the Pdgfra TAD border, overlapping an intron of the *Pdgfra* gene. The Hi-C map demonstrated that new spatial contacts between Pdgfra and Kit TADs were not established in the Pdgfra Δ2k MEFs (Fig. 3b), which is further confirmed by the subtraction map (Fig. 3c). The persistent insulation could be explained by the remaining CBSs at the Kit side of the boundary. Thus, the Pdgfra Δ2k strain represented a case of a partial disruption between two TADs.

The Pdgfra Δ60k mouse strain carries a 60 kb deletion of the entire Pdgfra/Kit TAD border, that removes the whole *Pdgfra* coding sequence but promoter and the first exon. Though initially we intended to simultaneously introduce two separate deletions at the Pdgfra and Kit sides of the Pdgfra/Kit border, we considered the Pdgfra Δ60k

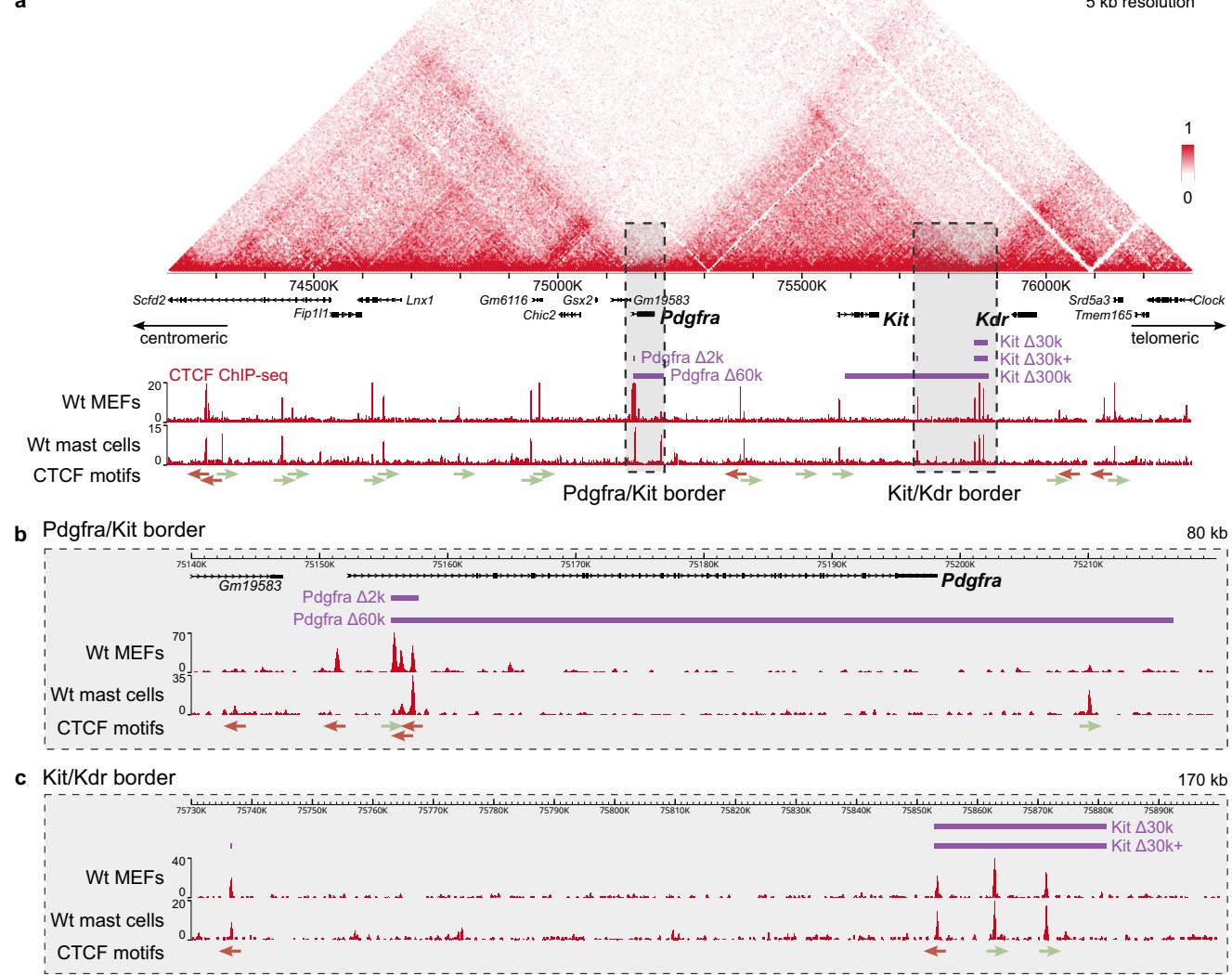

**Fig. 2 | Generated deletions to disrupt the 3D genome organization at the *Kit* locus. a** The studied TAD border regions illustrated on cHi-C from MEFs (gray rectangles). The CTCF binding sites are demonstrated by CTCF ChIP-seq from MEFs and mast cells. Arrows indicate forward (green) or reverse (red) orientation of a CTCF binding site. The deletions introduced to the borders are represented by purple rectangles: at the Pdgfra/Kit border (Pdgfra Δ2k, Pdgfra Δ60k) or the Kit/Kdr border (Kit Δ30k, Kit Δ30k + , Kit Δ300k). Magnification of the CTCF ChIP-seq data at the (**b**) the Pdgfra/Kit and (**c**) the Kit/Kdr provides a detailed view of the borders. Colour bar reflects the interaction counts.

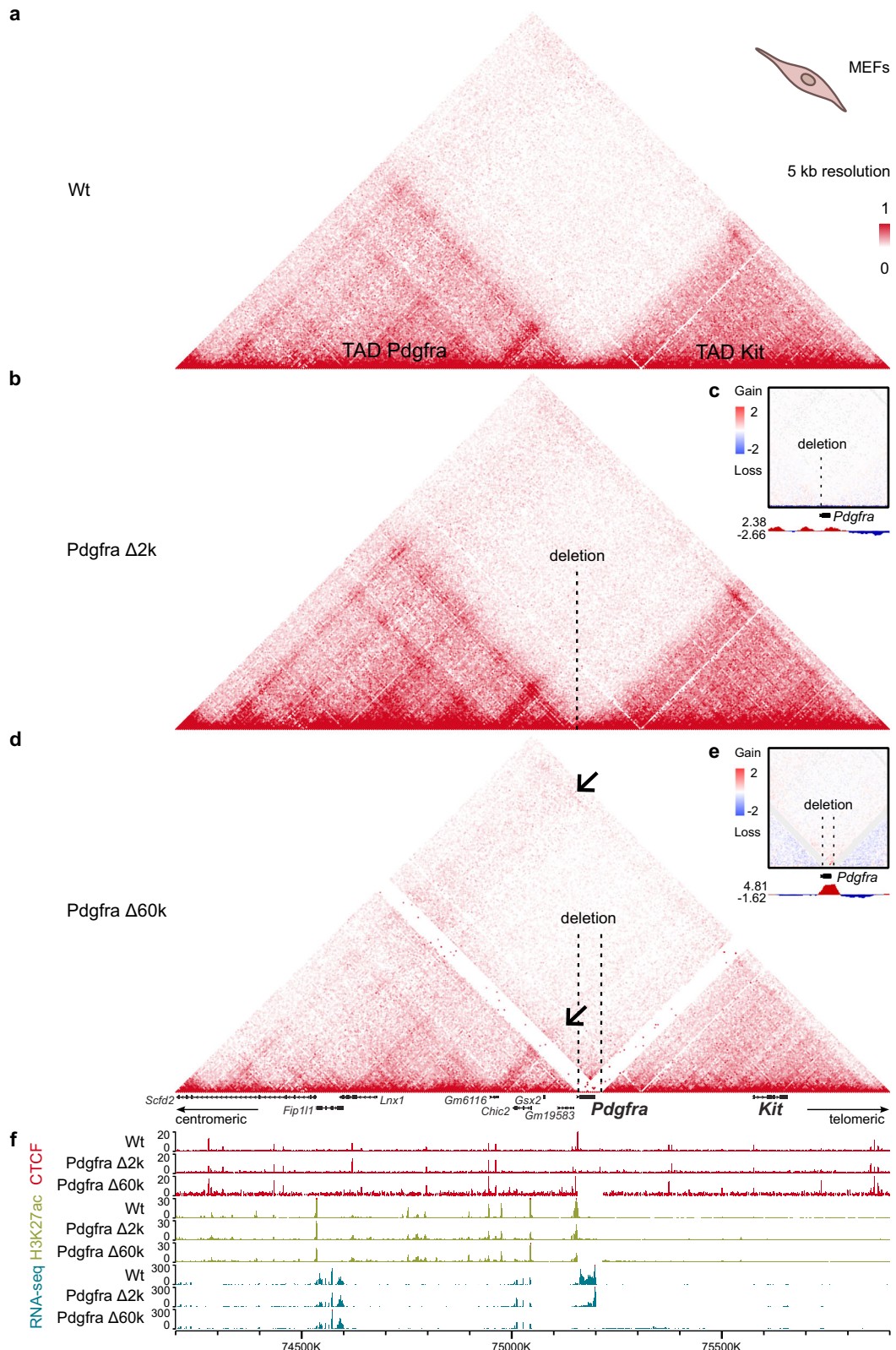

**Fig. 3 | 3D chromatin structure of the *Kit* locus in MEFs.** CBSs deletions Pdgfra Δ2k and Pdgfra Δ60k were insufficient to disrupt the TADs insulation between *Pdgfra* and *Kit* in MEFs, as demonstrated by comparison of the wild-type (Wt) (**a**), Pdgfra Δ2k mutant (**b**) and Pdgfra Δ60k mutant (**d**) Hi-C heatmaps (5 kb resolution). Colour bar reflects the interaction counts. Ectopic contacts in Pdgfra Δ60k are indicated by arrows. Spatial contact enrichment between TADs is additionally visualized by subtraction maps (10 kb resolution) for Pdgfra Δ2k (**c**) and Pdgfra Δ60k (**e**) relative to Wt. Gain of contacts in mutants is indicated in red, and loss of contacts in blue. The tracks below the subtraction maps show the estimated contact enrichment. **f** RNA-seq, CTCF and H3K27ac ChIP-seq signals across the *Kit* locus.

strain valid for the research purpose despite the *Pdgfra* knockout. Since the promoter and enhancers of the *Pdgfra* gene remained intact, we expected its activity to be sufficient for an enhancer hijacking.

*Pdgfra* is crucial for an early development, and its homozygous deletion is lethal for mouse embryos by 15 embryonic day (E15)[37]. Consequently, the Pdgfra Δ60k strain was maintained in a heterozygous state. Nonetheless, we were able to obtain fibroblast cultures from the E13.5 homozygous embryos.

The effect of Pdgfra Δ60k deletion was more distinct, compared to Pdgfra Δ2k. The contacts between the most distal regions became more pronounced, as a line above the TADs (Fig. 3d, arrows). Overall, the deletion caused enrichment of spatial contacts between the Pdgfra and Kit TADs (Fig. 3e). However, the changes were marginal. We believe that the TADs remained largely separated, that could be explained by a compensatory mechanism of a neighboring CTCF peak. Upon the deletion of boundary CBSs, a TAD boundary can shift towards the position of the nearest intra-TAD CTCF site[38,39]. CTCF ChIP-seq of Pdgfra Δ60k (Fig. 3f) revealed an increased binding of CTCF at the motif, located at a 4 kb distance from the TAD border and contained within the retained *Pdgfra* promoter. We have performed differential peak analysis to computationally assess the significance of our observations (Supplementary Fig. 4). We have used HOMER software which utilizes all the peaks detected across the genome to calculate the differential enrichment between Wt and mutant datasets. The *Pdgfra* promoter CTCF peak appeared to be differentially enriched with the $p$ value < 0.05. This may account for the robustness of insulation between the TADs.

We further asked if the mutations changed the expression pattern. The RNA-seq analysis of the mutant and Wt MEFs revealed that within a ± 1 megabase range surrounding the deletion site, there was no significant change in the expression of any genes, with the exception of *Pdgfra* (Fig. 3f). This exception is attributed to the deletion removing most of the *Pdgfra* gene. We found that an intergenic region between *Pdgfra* and *Kit* is transcribed upon Pdgfra Δ60k deletion, starting from the intact *Pdgfra* promoter and continuing to an intergenic region where it stops (Supplementary Fig. 5). But for an enhancer hijacking, H3K27ac enriched regions upstream to *Pdgfra* remained insulated (Fig. 3f), and consequently there was no increase in *Kit* transcriptional activity in any of the mutant MEFs.

The obtained data shows that Pdgfra Δ60k deletion disrupted the Pdgfra/Kit TADs border with marginal enrichment of spatial contacts between the two TADs, which was insufficient for an ectopic *Kit* activation. We presume that an adjacent CTCF, located further from the border, played a compensatory role explaining the robustness of the border. Considering the pivotal developmental role of the *Pdgfra*, its positioning at the TAD boundary may offer regulatory advantages. The preservation of this boundary might require additional factors, such as compensatory effects of neighboring CBSs, to ensure its stability.

## Kit/Kdr boundary CBSs deletions result in the TADs fusion
The Kit/Kdr boundary is a joint between two adjacent TADs which contains three clustered CBSs, with two sites in forward and one in reverse orientation (Fig. 2a, c). The Kit Δ30k mouse strain carries a 30 kb deletion of the three CBSs at the Kit/Kdr border. Interestingly, the Kit Δ30k deletion resulted in a phenotype change with the mice coat color changing to brown relative to black Wt.

Analyzing the first ChIP-seq data obtained from the Kit Δ30k mast cells, we noticed a strong signal of CTCF enrichment within the Kit TAD, located -116 kb away from the boundary. To minimize any potential impact of this site in maintaining the TAD's structure, we generated a Kit Δ30k+ strain, carrying an additional 300 bp deletion. However we found that both deletions had a similar effect on 3D genome organization in mast cells (see Supplementary Fig. 6 for Kit Δ30k+ effects in mast cells). Thus, we focused on the Kit Δ30k.

We monitored the effects of the deletion on two cell types where *Kit* is actively transcribed – mast cells and melanocytes. *Kdr* is normally inactive in mast cells and is marginally expressed in melanocytes (Fig. 1a). The resulting cell populations were highly homogeneous, as revealed by flow cytometry (Supplementary Fig. 3b), and displayed the typical morphological features. Mast cells produced heparin and histamine containing granules that can be detected by toluidine blue staining (Fig. 4a). Melanocytes demonstrated a typical morphology with granules of melanin in cytoplasm (Fig. 4b). Highly homogeneous cell cultures are crucial to gain pure sequencing data.

Contact maps from the Wt mast cells denoted a nested substructure of the Kit TAD, with two self-interacting regions. *Kit* with its upstream enhancers are located within the centromeric half of the TAD and thus are insulated from the telomeric half (Fig. 4c). Kit Δ30k deletion disrupted the Kit/Kdr TAD boundary insulation in mast cells and led to a noticeable gain of inter-TAD contacts. Looping interactions connecting the outer boundaries of Kit and Kdr TADs were also established. However, we noted that contacts rewiring mainly involved the telomeric half of the Kit TAD, which merged with the Kdr TAD. The insulation inside the Kit TAD was not affected by the deletion, and a self-interacting region, located at the centromeric side, remained isolated. Thus, despite the TAD boundary disruption in mast cells, active *Kit* enhancers remained insulated from *Kdr*.

Our CTCF ChIP-seq data confirmed a successful deletion of CBSs. The analysis did not reveal any CTCF peaks between two self-interacting regions inside the Kit TAD (Fig. 4c), nor an emergence of a novel CTCF peak in mutant cells, as was shown for Pdgfra Δ60k. Hence, *Kit* enhancers are possibly restrained from the downstream genes by different mechanisms rather than CTCF restricted loop extrusion, for instance, by the *Kit* transcriptional activity.

We then compared the effects of the boundary CBSs deletion in mast cells and melanocytes. The inner structure of the Kit TAD was divided with *Kit* and its downstream enhancers being insulated from the centromeric part of the TAD (Fig. 4d). Kit Δ30k deletion resulted in extensive rewiring of chromatin contacts in melanocytes. cHi-C maps revealed loss of insulation at the Kit/Kdr boundary and concurrent merging of neighboring TADs. Notably, spatial contacts were established between Kdr TAD and the telomeric part of the Kit TAD, enabling *Kit* enhancers to interact with the *Kdr* promoter.

The establishment of inter-TAD interactions in mast cells and melanocytes was further confirmed by the subtraction maps of mutant and Wt cells with simulated deletions (Fig. 4e, f). This approach was used to estimate the contribution of distance decrease on the rewiring of chromatin contacts and is discussed in more detail in the Supplementary Notes and demonstrated in Supplementary Fig. 7.

Thus, in both cell types, removal of boundary CTCFs resulted in loss of inter-TAD insulation and extensive interactions across the TADs boundary. At the same time, the insulation between two self-interacting regions within the Kit TAD was preserved. Our results imply that this insulation is provided by the epigenetic environment, in particular, by the relative position of enhancers and actively transcribed *Kit* within the TAD.

## The Kit/Kdr TADs fusion provides tissue-specific ectopic gene activation
We then asked whether the reorganization of contacts observed in the Kit Δ30k and Kit Δ30k+ mutant mice was functional and could trigger a transcriptional outcome. We noted that Kit Δ30k deletion was associated with a specific phenotype in adult mice. Mutant mice had brownish coat color, compared to the black Wt animals (Fig. 5a).

In mutant mast cells RNAseq analysis revealed no transcriptional changes in the locus or among the adjacent genes (± 1 Mb). Of note, for mast cells data from another study[40] reveals that all the region from the *Kit* termination site to the closest active gene *SrdSa3* appears to be

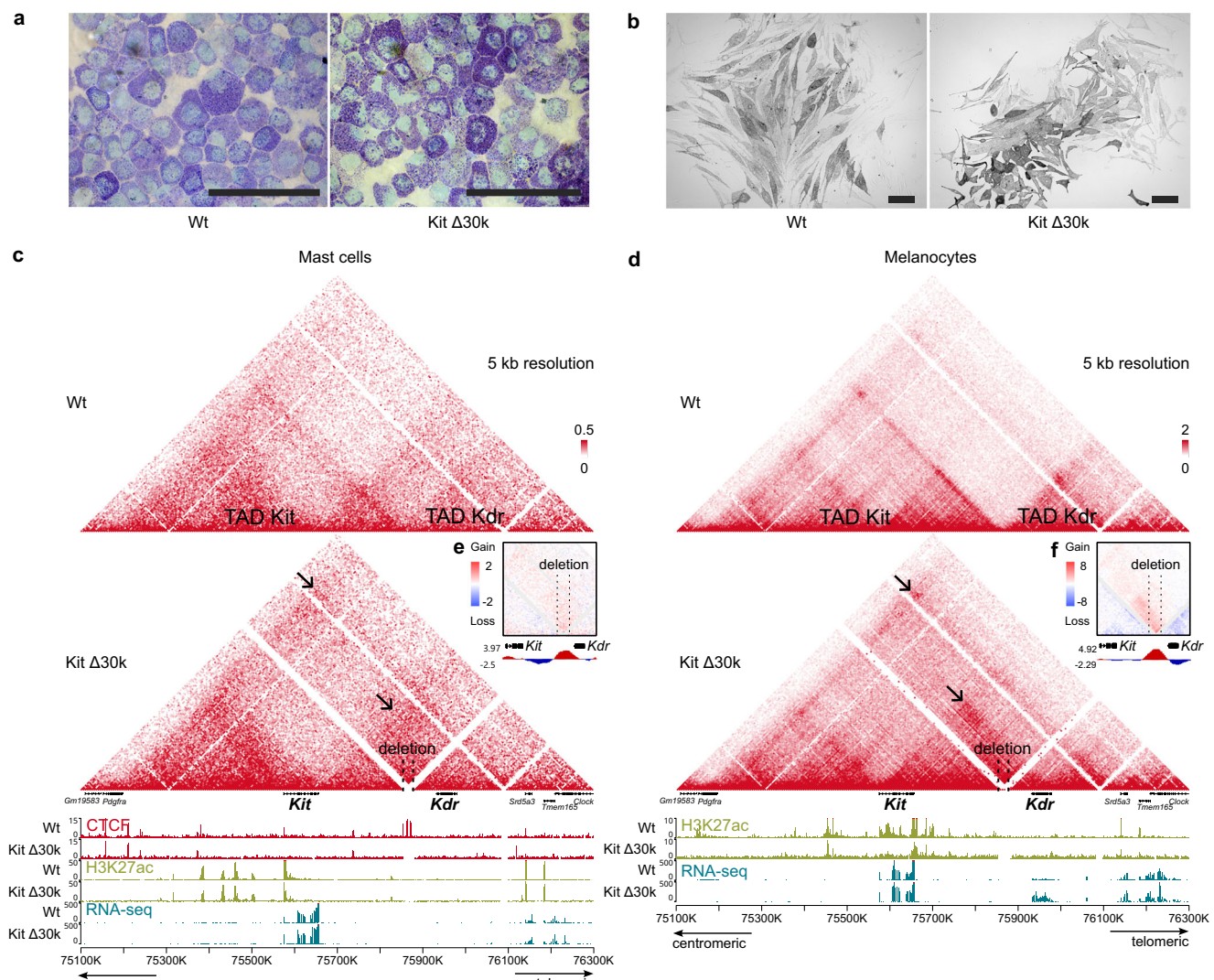

**Fig. 4 | Comparison of the TADs disruption at the *Kit* locus in mast cells and melanocytes. a** Cytospin preparations of Wt (left) and Kit Δ30k (right) mast cells after 4 weeks in culture. Histamine containing granules could be detected by toluidine blue staining. Scale bar: 100 um; **b** Primary Wt (left) and Kit Δ30k (right) melanocytes after 2.5 weeks in culture (without staining). Scale bar: 100 um. 3D genome organization and regulation context in Wt and Kit Δ30k mast cells (**c**) and melanocytes (**d**) are presented by Hi-C maps (5 kb resolution), CTCF and H3K27ac ChIP-seq, and RNA-seq. Colour bars reflect the interaction counts. Spatial contacts newly established after the TADs disruption are indicated by arrows. Subtraction maps demonstrate changes of spatial contact frequencies in mutant mast cells (**e**) and melanocytes (**f**) relative to the Wt. The tracks below the subtraction maps show the estimated contact enrichment, indicating loss of insulation in the mutant cells. Removal of the Kit/Kdr boundary resulted in loss of inter-TAD insulation and extensive interactions across the TADs boundary.

enriched in H3K27me3, suggesting a repressive state of *Kdr* (Supplementary Fig. 2a).

Remarkably, the situation was different in melanocytes. In Kit Δ30k cells, we detected a 6.7-fold increase in *Kdr* expression level compared to the Wt data (Fig. 5b). Notably, *Kdr* is normally expressed in melanocytes, although at a low level, suggesting an absence of repressive epigenetic state. Ectopic transcriptional activation observed in mutant melanocytes is in agreement with the redirection of the promoter-enhancer contacts revealed by cHi-C. We highlight that *Kdr* is the sole gene within a ±1 megabase range of the deletion that exhibited a significant change in expression.

Surprisingly, decrease of genomic distance between *Kit* and *Kdr* genes was also insufficient for *Kdr* activation in mast cells. We had performed transcriptome analysis on mast cells from a mouse strain carrying a 300 kb deletion at the Kit/Kdr TADs border, previously obtained in our laboratory[41] (Kit Δ300k, Fig. 2a). We found an abnormal transcription in Kit Δ300k that could be explained by transcription started from the *Kit* promoter, which remained intact after the

deletion, and continued to the closest termination site within *Kdr* (Fig. 5c). Of note, since *Kdr* locates on an opposing DNA strand to *Kit*, its transcript was antisense. The Kit Δ300k mice strain is heterozygous, therefore *Kit* was expressed from a normal allele. To compare effects of Kit Δ300k and Kit Δ30k deletions on 3D genome organization would be of interest, but unfortunately the Kit Δ300k is recessive lethal, so we were unable to obtain homozygous cells that are required to detect 3D genome changes.

In Kit Δ300k we did not detect a *Kdr* sense transcript, which potentially could be explained by the fact that an antisense transcript could inactivate a sense transcription through interference (see review[42]). But nor did we detect increased activity of genes downstream to *Kdr* which were unaffected by the antisense transcription. Thus, we believe that a high level of *Kit* transcription in mast cells could contribute to restriction of an enhancer hijacking rather than genomic distance.

Since *Kdr* expression is detected mostly in cardiac and endothelial cells, while *Kit* is expressed in a variety of cell types[43,44], we decided to

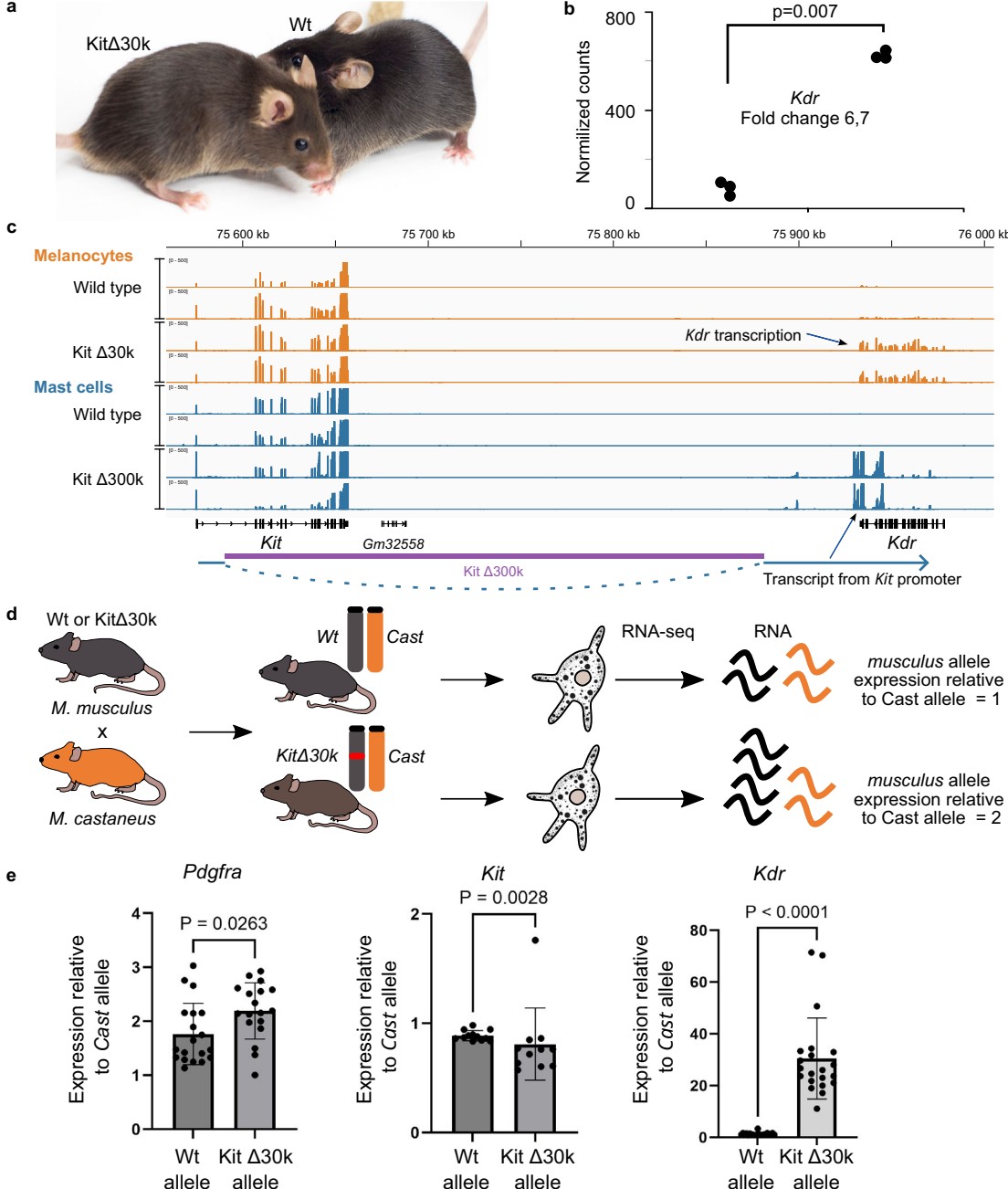

**Fig. 5 | Kit/Kdr TADs fusion induces *Kdr* activation in melanocytes. a** Coat color difference between brown Kit Δ30k (left) and black Wt (right) mice; **b** A plot of normalized RNA-seq reads counts in Wt and Kit Δ30k melanocytes indicates a 6.7-fold increase of *Kdr* transcription. Dots represent individual replicates. p-value was estimated with DESeq2, using the two-sided Wald test; **c** In mast cells Kit Δ300k deletion caused an antisense *Kdr* transcription; **d** Crossing scheme used to obtain *Mus musculus* (black) x *Mus castaneus* (orange) hybrids and further RNA-seq analysis; **e** Expression relative to a *M. castaneus* (*Cast*) allele. To evaluate an allelic activity, reads with *M. castaneus*-specific SNPs were scored for *Pdgfra* (*n* = 19 SNPs), *Kit* (*n* = 12 SNPs) and *Kdr* (*n* = 21 SNPs). Data are presented as mean values ± SD. *p* value was estimated using two-tailed Mann–Whitney test, NS – not significant. Source data are provided as a Source Data file.

find out if *Kdr* activation occurred in any other tissue due to our manipulations with the 3D genome structure. To this end we performed immunostaining in E15.5 embryos and looked for double-positive cells, expressing both *Kit* and *Kdr*. We focused specifically on somites, skin, eyes, vibrissae hair bulge, liver and yolk sac, attempting to find an aberrant Kdr signal in Kit-positive cells. Numerous double-positive cells were found in the skin of Kit Δ30k embryos (Supplementary Fig. 8). These cells were located at the epidermal layer and displayed branched, dendritic morphology. Kit-positive cells with identical shape in the skin of Wt embryos were negative for Kdr signal. Kit-expressing cell types, residing in the mouse fetal skin, are

melanocytes and mast cells, although the latter exclusively locate to the dermis[45]. Thus, the location of Kit-Kdr double-positive cells in the epidermal layer and their dendritic morphology suggests that these are melanocytes[43,46].

Notably, Kit Δ30k deletion could lead to *Kit* activation in *Kdr*-expressing endothelial cells, but we did not detect such activation, at least immunohistochemically. It can be assumed that in endotheliocytes, Kit Δ30k deletion did not cause adoption of *Kdr* enhancers by *Kit*.

As we mentioned earlier, we did not find any genes within a ± 1 mb range from the deleted Kit/Kdr TADs boundary that might have

significantly changed in expression. However, it can be hypothesized that in other cell types, the removal of the TAD boundary may affect more distant genes. Systematically testing this is challenging. Nevertheless, changes in the activity of genes that could produce a clearly visible phenotype are testable. We hypothesized that the key regulator of circadian rhythm, the *Clock* gene, located approximately 400 kb away (Supplementary Fig. 9), would be a suitable candidate for such a study. Alterations in gene activity might then influence the daily activities of animals. To test this hypothesis, we assessed changes in locomotor activity, sleep, and food and water intake in Kit Δ30k and Wt animals (Supplementary Fig. 9). We found no significant differences in these parameters, this leads us to tentatively suggest that the removal of the nearby specific TAD boundary might not significantly impact the regulation of *Clock* gene activity.

**Mus musculus x Mus castaneus breeding further confirmed ectopic *Kdr* activation to result from boundary CBSs deletions**
Since the melanocyte in vitro differentiation takes 4.5–7 weeks, it can be assumed that the cells obtained from Wt and Kit Δ30k mice could be exposed to the influence of confounding effects (cultivation conditions, trans-acting factors, differentiation state, etc.). Therefore, it is possible that increased *Kdr* expression does not reflect the difference in gene activity on the chromosome with and without deletion, but is only a consequence of different transcriptional status of cells. To test the hypothesis, we crossed Kit Δ30k and Wt *M. musculus* with *Mus castaneus* (Fig. 5d). Since almost all genes carry SNPs that distinguish *M. musculus* from *M. castaneus* (including *Pdgfra*, *Kit*, and *Kdr* genes), we were able to evaluate the activity of the genes using the reference *M. castaneus* allele as a normalization factor. Since both alleles are in the same cell, this assessment does not depend on transcriptional status of cells. We isolated primary cells from the epidermis of the hybrids, differentiated them, performed RNA-seq in melanocytes, and calculated the read counts from one of the two species based on the SNP analyses. Then, we used read counts from the *M. castaneus* allele as a normalization factor and compared the level of *Pdgfra*, *Kit* and *Kdr* expression between Wt and Kit Δ30k alleles. The level of *Kdr* expression was significantly lower in the control experiment with *M. castaneus* x *M. musculus* Wt hybrids, with both alleles being equally active (Fig. 5e). Thus, ectopic *Kdr* activation in melanocytes is truly caused by a deletion of the TAD border between the *Kit* and *Kdr* genes, and not by some confounding effects. Surprisingly, our analysis also revealed a downregulation of *Kit* in the KitΔ30k allele, which was modest and yet significant. This decreased *Kit* activity could be a result of the partial *Kit* enhancer retargeting[47,48].

Thus, the deletions of the boundary CBSs at the Kit/Kdr TADs border induced tissue-specific effects (Fig. 6). For mast cells, despite the weakened insulation between two TADs, the *Kit* enhancer region did not form new contacts with downstream promoters and *Kdr* remained inactive. In melanocytes, TAD boundary deletion induced the rewiring of *Kit* enhancers, which triggered *Kdr* misexpression and phenotype changes. We propose that the individual epigenetic characteristics of the locus, e.g., gene activity and enhancer localisation within the TAD, determine the tissue-specific response to a boundary deletion.

## Discussion

We found that TADs contribution to gene regulation depends on tissue-specific features at the *Kit* locus. The TAD border deletion caused fusion of the Kit and Kdr TADs in mast cells and melanocytes. Whereas an ectopic activation of *Kdr* occurred only in melanocytes. Several studies demonstrated that whether border CBSs deletion causes an ectopic gene activation largely depends on the locus, since the deletions provide inconsistent results (Supplementary Table 1). For the first time we provide comparison of the same locus at different regulatory landscapes in several pure primary cell cultures. Our results imply that TADs may unite different regulatory features, thus their function is not limited to an integrity of CBSs border, as was previously assumed.

### Deletion of a border between two TADs causes their fusion
We examined the impact of TADs disruptions in primary cell cultures, as opposed to an averaged data from tissues largely used before[4,13,49–51]. This approach enabled us to obtain high-quality functional genomic data. The data provided a view on chromatin interactions at different regulatory landscapes of the *Kit* locus. Three adjacent TADs encompass the locus, that is Pdgfra, Kit and Kdr TADs.

We found that removal of border CBSs resulted in TADs fusion, though architectural changes varied from a marginal increase of inter-TAD contacts across the Pdgfra/Kit border to the full contacts enrichment across the Kit/Kdr border.

We assume the CTCF site at the *Pdgfra* promoter, which remained intact after the Pdgfra Δ60k deletion, to account for an incomplete loss of an insulation between Pdgfra and Kit TADs. Though an insulating strength of an individual CBS is not correlated with its CTCF occupancy[52], we revealed the CTCF binding at the intact site to increase in Pdgfra Δ60k. It is in agreement with the fact that CBSs might be dispensable to maintain a TAD border, and neighboring CTCF might compensate for deletion of border CBSs[14,39,53]. It would be of interest to analyze how the deletion of the CBSs at the *Pdgfra* promoter affects 3D genome organization and gene regulation. But its deletion requires the deletion of the entire *Pdgfra* promoter. Loss of the target promoter was shown to retarget enhancer activity to the non-target

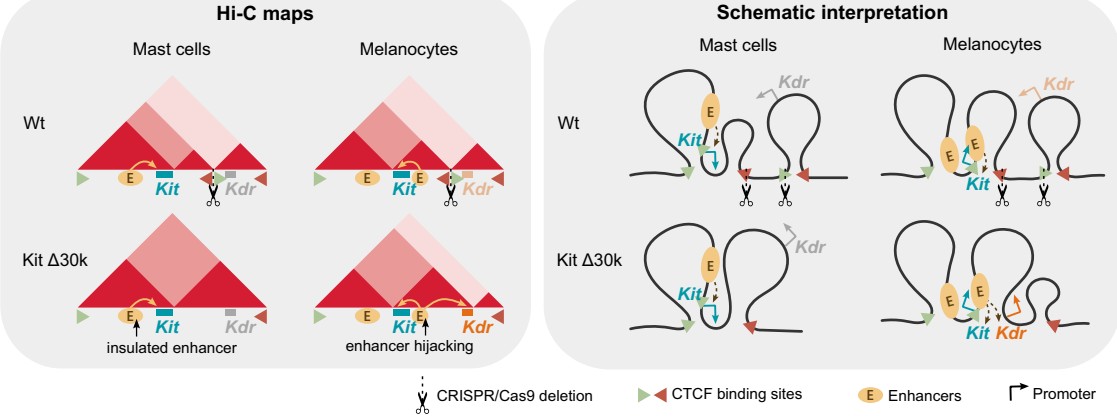

**Fig. 6 | Graphic summary of the study.** Disruption of the CTCF binding sites at the border between Kit and Kdr TADs results in *Kdr* activation in melanocytes. We suggest that it is due to the difference in regulatory context between mast cells in melanocytes, specifically localisation of enhancers.

genes located in the same TAD[54]. Therefore, it would be difficult to distinguish the effects of boundary deletion from the chromatin rewiring induced by the promoter loss. For this reason, we believe that deleting the CBSs at the *Pdgfra* promoter would trigger more extensive perturbations than just boundary deletion.

Another potential explanation for a robustness of the Pdgfra TAD to the introduced deletions is *Pdgfra* gene localisation at the border. Transcription from the *Pdgfra* promoter located near the TAD border − transcripts being the *Pdgfra* gene at the Pdgfra Δ2k and an intergenic region at the Pdgfra Δ60k − could strengthen the insulation. It was proposed that transcribed genic and non-genic boundaries are associated with stronger insulation due to boundary RNAs facilitating CTCF recruitment, with highly transcribed boundaries being the most CTCF-enriched[55]. But in case of Pdgfra Δ60k, since the level of intergenic mRNA in the Pdgfra Δ60k mutant is low, we would suspect the impact of this transcript on the CTCF binding to be rather weak. Thus, the Pdgfra TAD border was strengthened by a compensatory role of a neighboring CBSs or transcription at the border, and Pdgfra and Kit TADs remained largely separate. Since an enhancer influence on the promoter is constrained by TADs borders[56], a few newly established spatial interactions were insufficient for *Kit* activation in MEFs.

Since the distribution of genomic contacts is determined by a distance between loci, we emphasize that studying 3D genome organization upon deletions, especially large deletions, requires elimination of decrease in genomic distance due to deletions impact on the rewiring of chromatin interactions. To tackle the problem we utilized the C-InterSecture algorithm.

A limitation of our study is that we provide an analysis of the Pdgfra/Kit border in only one cell type. Though comparing an effect in MEFs with other cell types would be of interest, as an example glia has the same expression profile of *Pdgfra* and *Kit*, we believe that we would encounter the same problem of border robustness due to mechanisms discussed above. An additional constraint we were faced with is the TAD border disruption resulting in *Pdgfra* knockout mice, which is lethal in homozygous state, but disrupting the borders simultaneously and locally while preserving the integrity of *Pdgfra* poses a challenging task.

Of note, there is another mouse strain carrying the deletion of *Pdgfra* gene − *Patch* mutation. Interestingly, it causes an ectopic *Kit* activation in somites and lateral mesenchyme and exhibits a distinct pigment phenotype − white abdomen[57–59]. An analysis of changes in genome and spatial organization of the *Patch* mutant mice is yet to be performed. We suspect that the mutation covers a larger portion of the genome, enabling TADs fusion and consequently enhancer hijacking. Considering the substantial impact of deletions at the Pdgfra/Kit TADs border on mice development, we stipulate that the mechanisms we discussed, which impart robustness to this border, might be of great importance.

Concerning the Kit/Kdr border, the removal of CBSs at the TADs border led to loss of insulation, consequently resulting in the fusion of TADs. Herewith TADs fusion was different depending on the insulation pattern within the Kit TAD in mast cells or melanocytes. We discuss the result in more detail in the next section.

## *Kit* transcription defines tissue-specific characteristics of the TAD

Genome organization of the Kit TAD varies between the examined cell types. The TAD demonstrated homogeneous spatial contacts in MEFs, whereas in mast cells and melanocytes the Kit TAD was divided into two distinct self-interacting regions. We noted that partition of the TAD occurred at the *Kit* gene (which is the only gene at the TAD), and insulation point differed locating at the *Kit* promoter in melanocytes and the *Kit* terminator in mast cells. We presume *Kit* transcription to provide the insulation in the TAD, because *Kit* is inactive and an insulation is absent in MEFs. It is in agreement with the study revealing that

transcription modulates chromosome folding during mouse thymocyte maturation with single activated genes forming an insulated region within TADs[60]. Models of cohesin loading and transcription at different states suggested RNA polymerase II (RNAP) to act as a barrier to loop extrusion[29]. Transcribing RNAP may disrupt chromatin interactions by displacing cohesin from CTCF sites[61,62] and relocating it over long distances along DNA[63]. Moreover, upon depletion of RNAP many enhancer-promoter loops are lost and new loops are established at the CTCF sites, which were restricted by RNAP[64].

It is of great interest that insulation within the Kit TAD at the promoter or terminator of *Kit* depends on the cell type. Transcription can participate in the formation of nested structures within TADs, which are established independently of CTCF-cohesin loop extrusion mechanism[65]. But data from Wt MEFs and mast cells revealed a CBS within the *Kit* promoter, which could account for the insulation within the TAD. Though *Kit* is expressed in both cell types, the expression level in mast cells is ten times higher than in melanocytes. High level of transcription is assumed to antagonize CTCF-cohesin loop extrusion[66]. Cohesin was revealed to accumulate at 3′-end of highly active genes in an absence of CBS or an efficient cohesin release by WAPL[63,67]. This implies that cohesin could be shifted from the CBS at the *Kit* promoter to its terminator resulting in 3D genome organization revealed by our Hi-C maps in mast cells, while this effect was absent in melanocytes. Thus, a level of *Kit* expression could influence tissue-specific features of the spatial genome organization at the TAD.

## TAD disruption results in *Kdr* activation in melanocytes, but not in mast cells

How TADs integrity affects gene regulation is a matter of great interest, given TADs are assumed to establish enhancer-promoter interactions.

Our transcriptome analysis of Kit Δ30k cells revealed an ectopic activation of *Kdr* in mutant melanocytes, whereas *Kdr* remained silent in mutant mast cells. The plausible explanations for the cell type specific enhancers adoption by *Kdr* are different transcription levels of *Kit* and an epigenetic status of *Kdr* in mast cells and melanocytes.

Genomic distance is potentially another factor to influence an enhancer adoption. We were able to assess its contribution to *Kdr* insulation in mast cells upon a larger deletion at the Kit/Kdr TADs border (300 kb against 30 kb). Transcriptome analysis revealed the lack of *Kdr* transcription. Though we detected an antisense transcript including *Kdr* gene, which could potentially inactivate a sense transcription through interference (see review[42]). But nor did we detect increased activity of genes downstream to *Kdr* which were unaffected by an antisense transcription. Thus, we conclude that other factors rather than genomic distance restricted enhancer hijacking in mast cells.

We found *Kdr* to be expressed at a low level in Wt melanocytes, and silent in Wt mast cells. Taking into account *Kdr* enrichment of a repressive epigenetic mark H3K27me3 in mast cells, it is possible that the repressive state of *Kdr* makes it inaccessible to an enhancer hijacking. Our Hi-C maps revealed that spatial interactions between *Kit* and *Kdr* were not established in mast cells, and the active enhancers remained insulated from *Kdr*. However, whether the repressive state of *Kdr* could influence the spatial insulation is unclear, as well as the causation between gene activity and epigenetic state. We consider the high level of *Kit* transcription in mast cells (which is tenfold higher relative to melanocytes) to be a more likely explanation for the insulation, since RNAPII was revealed to act as a barrier for cohesin-mediated loop extrusion[29].

We additionally confirmed *Kdr* activation in melanocytes to result from the Kit Δ30k deletion by breeding Kit Δ30k *M. musculus* with Wt *M. castaneus*, showing that the vast majority of the *Kdr* transcripts (98%) originated from a *M. musculus* allele. The observed difference corresponds to the 3D architecture of the locus upon Kit Δ30k deletion. Of note, we did not detect a reverse situation of *Kit* activation by

active enhancers of *Kdr* in endothelial cells. Thus, cell type-specific features of the locus, such as localization of H3K27ac-marked active enhancers, underlie the TADs function, at least in case of the *Kit* locus.

Our *M. musculus* x *M. castaneus* breeding experiment revealed downregulation of *Kit* in the Kit Δ30k melanocytes. Although significant, the difference in relative *Kit* expression level was slight (Wt/*castaneus* 0.88 → Kit Δ30k/*castaneus* 0.75), compared to the dramatic *Kdr* upregulation (Wt/*castaneus* 0.99 → Kit Δ30k/*castaneus* 49.71). Downregulation of gene activity due to an enhancer retargeting may reflect the promoter competition phenomenon. There is some evidence of the dilution of the enhancer activity among multiple genes due to the TAD boundary disruption, which leads to the downregulation of target genes (see the review[68]). Here we might see a similar mechanism.

Cell type-specific effect of TADs disruption on gene expression was also demonstrated in comparisons of induced pluripotent stem cells (iPSCs) and iPSC-derived cardiomyocytes[69], ESCs and motor neurons[38]. In addition, strength of TADs border may vary between cell types, as was shown for mESCs and MEFs[53]. Though given a generally more open chromatin state of stem cells relative to differentiated cells[70], the advantage of our study is providing evidence of the effects in primary cell cultures from animals with deletions.

Studies concerning the influence of TADs disruption on expression suggest that regulation within TADs may largely depend on other factors rather than on the integrity of a TAD border only (Supplementary Table 1). We provide evidence of cell-type specific features of the *Kit* locus to affect a TAD's role insulating an enhancer influence on a non-target promoter.

### Phenotypic consequences of the Kit TAD disruption

In our study we observed a change in the mice phenotype as a result of the CBSs deletions at the border between Kit and Kdr TADs: the coat of the Kit Δ30k mice were lighter relative to the wild-type mice. It is well-known that mutations in *Kit* are implicated in the white-spotting phenotype in mice[21,71,72], rabbits[73,74], cats[75], dogs[22], horses[76] and other animals. The mechanism for this phenotype is an impaired melanocyte development and migration due to the defective Kit receptor signaling. An assumption that *Kit* downregulation detected in melanocytes could explain a paler phenotype in Kit Δ30k mice can be made, although it faces an important contradiction. *Kit* mutations described in earlier studies result in a total absence of pigmentation, either locally or all over the body, while Kit Δ30k mice have lighter coloration, which is an uncharacteristic manifestation of *Kit* mutations. However, it is obvious that coloration is an essential object for natural selection, so even such an almost invisible change in the laboratory could be eliminated in natural conditions.

Mutations leading to TADs fusion are under negative selection[77], thus, the evolutionary conservatism of TADs may be a consequence of their importance for maintaining a certain pattern of gene activity. Whereas a change in the structure of TADs can lead to the emergence of a new pattern of gene activity, which may be an evolutionary innovation. This was remarkably demonstrated by the example of moles[78]. One of the adaptations of moles to an underground lifestyle is to increase the concentration of testosterone in females. An aberrant pattern of *Fgf9* gene activity ensures the tolerance of female oogenesis to the male hormone. To change *Fgf9* activity, a chromosomal rearrangement, affecting the distal part of the *Fgf9* TAD, occurred in moles. Given TADs role in insulating a regulatory pattern, they ensure an integration of a newly established gene or a change in regulation to occur without huge alterations in neighborhood[79]. Thus, the regulatory context provided by TADs is an important object for the evolution of organisms.

### Perspective

Our findings indicate that the consequences of TAD border deletions may largely depend on the specific characteristics of an individual locus within a particular cell-type context. We revealed that an epigenetic landscape in the *Kit* locus, especially an active transcription and an enhancer localization, accounts for a different insulating role of the *Kit* TAD in mast cells and melanocytes. We propose that future research should dive into understanding the interplay and relative contributions of these features in shaping the insulation role of TADs.

## Methods

### Animals

Mouse Kit Δ30k line was established from a chimeric animal generated after mouse embryonic stem cells (mESCs) injection as described earlier[80]. Briefly, mESCs were transfected with CRISPR/Cas9 plasmids aimed at a border region between Kit and Kdr TADs (see gRNA in Supplementary Table 2). With PCR genotyping we selected a homozygous clone with Kit Δ30k deletion (see primers in Supplementary Table 2). Modified mESCs were injected into recipient blastocysts. Chimeric founders (F0) were crossed with wild-type (Wt) C57BL/6 J animals to generate heterozygous animals (F1). After 8 rounds of backcrossing to C57BL/6 J animals, a homozygous Kit Δ30k line was established.

All other mouse lines in this study were generated through cytoplasmic injection of mRNA Cas9 and gRNAs (Supplementary Table 2), as described in Korablev et al., 2019. For generation Pdgfra Δ2k and Pdgfra Δ60k lines CRISPR components were injected in C57BL/6 J zygotes. For generation Kit Δ30k+ line CRISPR components were injected in Kit Δ30k zygotes.

The daily activity of animals was assessed using a Phenomaster. For this purpose, we compared two groups of 12-week-old males from the Kit Δ30k and C57BL/6 lines. The animals were trained to use the drinking bowls and feeders over two consecutive days. Following this, they were isolated in the Phenomaster cages, where their total locomotor activity, sleep duration, number of sleep episodes, and water and food consumption were recorded over a period of 72 h.

Animals were kept in a standard environment at 24 °C temperature, 40–50% relative air humidity and 14 h light/10 h dark–light-cycle. Food and water were available ad libitum. At the end of experiments, remaining animals were euthanized by $CO_2$. All the procedures and technical manipulations with animals were in compliance with the European Communities Council Directive of 24 November 1986 (86/609/EEC) and approved by the Bioethical Committee at the Institute of Cytology and Genetics (Permission N45 from 16 November 2018).

### Cell cultures

**Mouse embryonic fibroblasts (MEFs).** MEFs were obtained from E13.5 as described in[81]. MEFs were cultured at 37 °C under 5% $CO_2$ in Dulbecco's Modified Eagle Medium (DMEM) (Thermo Fisher Scientific, 12800082), supplemented with 10% FBS (Capricorn Scientific, FBS-11A), 1x penicillin & streptomycin 10x (Capricorn Scientific, PS-B), 1x GlutaMax-I 100× (Thermo Fisher Scientific, 35050061). For subculture cells were rinsed with 1 × PBS and detached using 0.25% trypsin-EDTA at 37 °C for 3 min. Cells were typically split every 2–3 days at a 1:2 ratio. Flow cytometry showed more than 80% *Pdgfra* positive cells (Supplementary Fig. 3b).

**Melanocyte cell culture.** Melanocytes from Wt and Kit Δ30k mice were obtained according to the Murphy et al., protocol[82] with modifications. In brief, 4-day-old pups were decapitated, the skin was separated, rinsed in 10x penicillin/streptomycin solution (Capricorn Scientific, PS-B), and incubated overnight in 0.25% trypsin/DMEM (Capricorn Scientific, TRY-2B). Next day, blood, adipose and muscle tissues were removed, epidermal and dermal layers were disassociated. Epidermis was chopped with scissors in 0.25% trypsin/PBS solution, and additionally incubated in 0.25% trypsin for 10–15 min at 37 °C. Then, the cell suspension was filtered through a nylon mesh to remove large tissue segments, cells were pelleted by centrifugation

and pre-plated for 25–30 min. Finally, unattached cells were transferred to the plastic surface 6-well plate, covered with 0.1% Gelatin from porcine skin type B (Sigma, G9391), and cultivated in Melanocyte growth medium (PromoCell, C-39410), containing geneticin 100 ng/ml (G418) (Thermo Fisher Scientific, 11811023), for 48 h. On day 3, the medium was replaced with a fresh one, without G418. At that step individual melanocytes could be clearly visualized. In 10–14 days, the majority of the cell population represented granulated melanocytes. The cells were grown for 4–7 weeks until the required cell quantity was reached and more than 80% of cells were *Kit* positive (Supplementary Fig. 3b).

**Mast cell culture**. Mast cells were differentiated from hematopoietic cells derived from bone marrow according to the Vukman et al., protocol[83]. Cells were cultured for 8 to 10 passages in Iscove's Modified Dulbecco's Medium (IMDM) (Thermo Fisher Scientific, 12200069) supplemented with 10% FBS (Thermo Fisher Scientific, 16141002), mrSCF(10 ng/ml) (Biolegend, 579702) and mrIL-3 (10 ng/ml) (Biolegend, 575502). Mast cells were also stained by toluidine blue (Biovitrum, 07-002) − a basic thiazine metachromatic dye that stains nuclei blue and has a high affinity for acidic granules, in accordance with the manufacturer's recommendations.

According to the flow cytometry analysis of *Kit* and *FcεR1* expression, at the final passages mast cells accounted for ~90% of the total cell population (Supplementary Fig. 3b). For the Hi-C experiment *Kit* positive cells were magnetically separated (Miltenyi Biotec, 130-097-146).

**Flow cytometry (FC)**
FC was performed as described in[84]. A total of ~0.5-1 * 10^6 cells/ml were harvested and washed in PBS. Antibody and cell suspension were mixed in an ice-cold FC buffer (PBS, 10% FBS) in a ratio 1:300 and incubated on ice for 1 h. After incubation cells were pelleted by spinning at 300×g for 10 min. The pellet was washed in the FC buffer twice to remove unnecessary antibodies. The pellet was resuspended in a 500 μL buffer and filtered through nylon mesh to remove cell conglomerates. A negative control was prepared as a cell suspension washed in PBS without immunostaining. To exclude cells with compromised membranes cell suspension was incubated with 7-aminoactinomycin D (7-AAD) (Thermo Fisher Scientific, A1310) in a ratio 1:1000 for at least 20 min, after which the cell pellet was washed as described above. The prepared cell suspensions were analysed using BD FC Aria III (BD Biosciences). The antibodies were from BioLegend: APC/Cyanine7 anti-mouse CD117, c-Kit (#135136); APC anti-mouse CD117, c-Kit (#105812); PE anti-mouse CD140a, Pdgfra (#135905); FITC anti-mouse FcεRIα (#134306).

**ChIP-seq**
The anti-histone H3 (acetyl K27) ChIP-seq and input libraries were prepared for each cell type as described in Pekowska et al.[85] with slight modifications. The anti-CTCF ChIP-seq was performed in MEFs and mast cells. Briefly, ~ 5 * 10^6 cells were harvested per experiment and crosslinked with 1% formaldehyde at RT for 10 min. Crosslinking was quenched with 125 mM glycine. Cells were lysed on ice in 0.5 ml lysis buffer (10 mM Tris pH 7.5, 1 mM EDTA, 0.4–1% SDS, 0.1% sodium deoxycholate, 1% Triton X-100, Complete Mini EDTA-free protease inhibitors (Roche, 11836170001)). Lysed chromatin was fragmented using the BANDELIN SONOPULSE until reaching a fragment size of 150–500 base pairs (5–10 cycles, 30 s/60 s on/off, 60% amplitude, 4°C). SDS concentration and sonication cycles were adjusted according to a cell type and shearing efficiency. Lysates were clarified by 16,000 g centrifugation for 10–15 min at 4 °C and diluted with 1–1.5 ml of lysis buffer without SDS to reduce SDS concentration before the immunoprecipitation step (~0.17% SDS final). Chromatin was pre-cleared with 20 μl Protein A magnetic beads (New England Biolabs, S1425S) for 1.5 h

and 100 μL of the solution was saved as an input control. During this time, a 40 μl aliquot of Protein A beads was washed with PBS and rotated with the respective antibody at 4°C (1.5 − 5 μg antibody). Protein-DNA complexes were then immunoprecipitated overnight at 4 °C with rotation. The beads were washed at 4°C in the following buffers: lysis buffer (0.1% SDS, twice), lysis buffer containing 0.5 M NaCl (twice), LiCl buffer (0.25 M LiCl, 0.5% IGEPAL-630, 0.5% sodium deoxycholate, twice), TE (pH 8.0) plus 0.2% Triton X-100 (once), and TE (pH 8.0, once). Crosslinks were reversed at 65°C overnight and DNA was purified with ChIP DNA Clean & Concentrator kit (Zymo Research, D5205). Sequencing libraries were prepared using the KAPA Hyper Prep kit. The anti-histone H3 (acetyl K27) (Abcam, ab4729) and anti-CTCF (Abcam, ab70303) antibodies have been successfully validated by Western Blot, dilution 1:3000 (Supplementary Fig. 10). The experiments were performed in two biological replicates.

**Capture Hi-C (cHi-C)**
Hi-C from two biological replicates of each cell type were performed as previously described in[86] with some modifications from[87]. A total of ~ 5 * 10^6 cells were crosslinked in 1–2% PFA/PBS buffer at RT. A cell pellet was resuspended in a lysis buffer (150 mM Tris-HCl pH 7.5, 140 mM NaCl, 0.5% NP-40, 1% Triton X-100, 1x Complete Mini EDTA-free protease inhibitors) and incubated for 15 min at 4°C followed by 15 min at RT. Extracted nuclei were spun down at 800 g then washed once in lysis buffer and once in NEBuffer3.1 (New England Biolabs, B703S). After centrifugation, nuclei pellet was resuspended in NEBuffer3.1 supplemented with 0.3% SDS, then incubated at 37 °C for 1 h followed by 10 min at 65 °C to open chromatin. 10% Triton X-100 was added to the reaction (1.8% final) to quench SDS and incubated at 37 °C for 30 min. Chromatin was digested overnight with 400U DpnII (New England Biolabs, R0543M). Following the inactivation of DpnII, the nuclei were spun down at 2500 g and resuspended in NEBuffer2.1. Biotin fill-in was performed with biotin-14-dATP (Invitrogen, 19524016) using Klenow DNA polymerase (SibEnzyme, E325) at 22 °C for 4 h. Samples were centrifuged and the pellet was resuspended in ligation mixture (1xT4 Ligation buffer (New England Biolabs, B0202S), 5% PEG, 1% Triton X-100, 0.1 mg/ml BSA and T4 ligase). Ligation was carried out in 1.2 mL volume overnight at 16 °C with mixing. The ligation products were reverse crosslinked at 65°C overnight with proteinase K (New England Biolabs, P8107S) and DNA was extracted with phenol-chloroform. Efficiency of DNA digestion and ligation was assessed with gel electrophoresis. Sonication was performed on the Covaris M220 system to obtain 200–400 bp fragments. DNA fragments were double size-selected using AMPure XP beads (Beckman Coulter, A63880). MyOne Streptavidin C1 magnetic beads (Invitrogen, 65001) were added to bind biotin-tagged fragments. Sequencing libraries were prepared using the KAPA Hyper Prep kit. The hybridization of libraries with RNA probes was performed according to the myBaits Manual v4.01 (Arbor Biosciences). Enrichment probes were designed over the region chr5:74,135,000-76,410,000.

**RNA-seq**
0.2*10^6–2*10^6 cells were harvested and washed twice with PBS. Total RNA was extracted using Aurum Total RNA mini kit (Bio-Rad, 7326820). Sample concentration was evaluated using Qubit RNA HS (Q32852, Invitrogen). Gel electrophoresis was used to verify the integrity of extracted RNA. Three replicates were generated for each cell type and genotype, respectively.

**Immunohistochemistry (IHC)**
E13.5-15.5 mice were fixed with 4% PFA solution in 1X PBS. Skin tissue sections from 4-day-old mice were harvested from head and back in 1X PBS and fixed in 4% PFA solution overnight on a roller shaker at 4 °C. Next day samples were washed in 1X PBS solution three times for 30 min. For tissue dehydration organs were sequentially incubated for

at least 24 h with a 15% and 30% sucrose solution in 1X PBS at 4 °C. Next, organs were embedded in Tissue-Tek O.C.T. Compound (Sakura Finetek, 4583) and frozen. Organ sections of 50 um thickness were prepared on MICROM HM 505 N cryostat (Microm) and immediately collected on SuperFrost™ slides (Thermo Fisher Scientific). Sections were washed with 1X PBS and incubated in blocking solution: 2% BSA (Sigma Aldrich, A2153), 0.2% Triton X-100 (Amresco, X100), 5% FBS (Capricom Scientific, FBS-11A). Primary antibodies anti-Kdr (R&D, AF644) and anti-Kit (CellSignaling, 3074) were diluted in blocking solution (1:100) and incubated overnight at slow speed on an orbital shaker at room temperature. Next, slices were washed with 1X PBS three times for 20 min. Slices were stained with secondary antibodies (Jackson Immuno Research, 705-165-147 and 711-545-152) in dilution 1:500 and DAPI diluted in 1X PBS for 2 h at room temperature. Slices were washed with 1X PBS three times for 20 min and completely dried out and finally were mounted with ProLong™ Diamond Antifade Mountant (Thermo Fisher Scientific, P36965). Immunofluorescence were visualized under confocal fluorescence microscope LSM 780 NLO (Zeiss) with ZEN software (Zeiss). Microscopic analysis was carried out at the Multiple-access Center for Microscopy of Biological Subjects of the Institute of Cytology and Genetics SB RAS.

### Computational methods

All computations except the RNA-seq data analysis were performed using nodes of Novosibirsk State University high-throughput computational cluster.

**cHi-C data processing.** The cHi-C sequencing data were processed using Juicer software Version 1.11[88] and mapped to the mm10 genome. Contact maps were generated from read pairs with MAPQ ≥ 30 and normalized using the VC_SQRT method. Maps at 5 kb resolution were further visualized via the Epigenome browser[89].

**TAD organization analysis.** The Hi-C maps for human (hg38), mice (mm10), rabbit (oryCun2), dog (canFam3), chicken (galGal5) were obtained from GSE167579[34]. The Hi-C map of african clawed frog (xenLae2) was obtained from[35].

The conservativity of the *Kit* locus was estimated by the C-InterSecture software[33]. For the cross-species comparison, we generated maps of synteny regions from the pairwise genome alignments. At first, the genomes of interest were aligned by LastZ Version 1.04.22[90]. Then, the given alignments were converted into net-files by KentUtils [https://github.com/ucscGenomeBrowser/kent] and transformed into the syntenic maps by the C-InterSecture script. Additionally, with a considerable part of the genomes being not included in any synteny blocks, we mechanically filled gaps (<1 Mb) splitting codirected synteny blocks. The resulting synteny maps were used to liftover Hi-C contacts between the genomes by a "balanced" model.

**Computation of the contacts enrichment.** The comparison of capture Hi-C of wild type and mutant was performed by the C-InterSecture software, too. The synteny maps were manually generated according to the coordinates of deletion. Then, the Hi-C contacts of wild type were liftovered on the mutant genome by a "balanced" model, as though the deletion effects on chromosome architecture only by changing of a genomic distance. Finally, the given contacts were liftovered from mutant genome on wild type by an "easy" model accounting only synteny between locus.

The difference between the based and liftovered Hi-C map was calculated thus:

Let i, j and k - is genome coordinate in bins. Then:

$V_{i,j}$ is the contact value between bins i and j.

$V_{i,j}^*$ is the liftovered contact value between bins i and j.

$D(k,f) = \overline{\log(V_{i,j}/V_{i,j}^*)}$ for $i \in [k-f,k), j \in (k, k+f], V_{i,j} \neq 0, V_{i,j}^* \neq 0$, and $f$ is distance in bins

Then, we performed a Z-score transformation of D(k,f) for all k and chosen f.

**ChIP-seq data processing.** Raw paired-end reads were preprocessed with Cutadapt tool to trim adapter sequences (-a and -A flags with Illumina adapter sequences)[91]. Read quality was assessed using FastQC. Coverage tracks (bigWig) were generated using AQUAS ChIP-Seq pipeline1 with TF or histone parameters [https://github.com/kundajelab/chipseq_pipeline].

To determine a CTCF motif orientation we used the FIMO tool Version 5.5.0 from MEME[92] with the matrices MA0139.1 and MA1929.1 from JASPAR[93].

To perform a differential ChIP-seq analysis we used getDifferentialPeaksReplicates.pl tool from HOMER software Version 4.11 collection of command line programs[94] with standard parameters using mouse genome of mm10 version and Tag directory. We utilized the makeTagDirectory.pl command from HOMER software Version 4.11 to generate a Tag directory. This was performed using the narrow peak files obtained from the AQUAS ChIP-Seq pipeline1 [https://github.com/kundajelab/chipseq_pipeline] for two biological replicates of both wild-type and mutant CTCF ChIP-seq experiments.

**RNA-seq data processing.** Sequencing data were uploaded to and analyzed on the Galaxy web platform usegalaxy.org[95]. Quality of sequencing data was controlled via FastQC Galaxy Version 0.73[96]. Reads were mapped to the mouse reference genome mm10 (GRCm38) with RNA STAR Galaxy Version 2.7.8a[97] (default parameter) using the Gencode main annotation file release M1 (NCBIM37). Mapped reads were counted using featureCounts Galaxy Version 2.0.1[98] (default parameter). Differential gene expression and normalized counts were calculated using DESeq2 Galaxy Version 2.11.40.7[99] (default parameter).

To evaluate alleles activity in hybrid melanocytes we used *Mus castaneus* SNPs identified by the Mouse Genomes Project consortium[100,101]. The identified number of SNPs inside the analyzed genes was: *Pdgfra* − 19, *Kit* − 12, *Kdr* − 21. For each SNP, we calculated the normalized activity of the Wt or Kit Δ30k allele relative to the *M. musculus* allele. In order to avoid the problem of division by zero, a pseudocount (1) was added to each SNP coverage value before normalization: Normalized allele expression $= \frac{M.musculus\ \text{SNP coverage}+1}{M.castaneus\ \text{SNP coverage}+1}$. p-value was estimated using Mann–Whitney U test.

### Statistics and reproducibility

Hi-C, RNA-seq, and ChIP-seq experiments were repeated at least in two independent replicates for all cell cultures. Staining with toluidine blue of mast cells was performed at least in three independent replicates and was reproducible.

### Reporting summary

Further information on research design is available in the Nature Portfolio Reporting Summary linked to this article.

## Data availability

The raw sequencing data have been deposited in the NCBI SRA database with the following accession number PRJNA838252. Processed data, including Hi-C contact maps, RNA-seq and ChIP-seq tracks are available at https://genedev.bionet.nsc.ru/ftp/by_Project/Kit_locus_GEO. The data used for other organisms are publicly available: fibroblasts[34] of human, mouse, rabbit, dog and chicken; African clawed frog fibroblasts[35]. Source data are provided with this paper.

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

## Acknowledgements

This work was supported with the budget project of Institute of Cytology and Genetics SB RAS (state project FWNR-2022-0019). Experiment with *M. castaneus* hybrids was supported by Russian Science Foundation grant #22-14-00247. Hi-C experiments sequencing was performed using equipment of the Novosibirsk State University, supported by the Ministry of Education and Science of Russian Federation, grant #FSUS-2024-0018. Illumina data analysis was supported by the strategic academic leadership program "Priority 2030" in Novosibirsk State University. Cell culturing was performed at the Collective Center of ICG SB RAS "Collection of Pluripotent Human and Mammalian Cell Cultures for Biological and Biomedical Research", project number FWNR-2022-0019 (https://ckp.icgen.ru/cells/; http://www.biores.cytogen.ru/brc_cells/collections/ICG_SB_RAS_CELL). The calculations were performed using computational resources of Computational Center of Novosibirsk State University.

## Author contributions

N.B. conceived and supervised the study; E.K., A.R., V.L. and A.Kh. performed Hi-C, ChIP-seq, RNA-seq experiments; M.N. performed Hi-C data analysis; P.B. performed ChIP-seq and RNA-seq data analysis; T.S. performed immunohistochemistry analysis; A.S. designed the genetic constructs; N.V.K. conducted the experiment to assess the circadian rhythms of the animals; A.Ko., G.K. and I.S. obtained genome–edited mice. All the authors contributed to the manuscript preparation.

## Competing interests

The authors declare no competing interests.

## Additional information

**Peer review information** : *Nature Communications* thanks Lila Allou, Martin Franke and the other, anonymous, reviewer(s) for their contribution to the peer review of this work. A peer review file is available.

