## [Peer Review File · Nature Communications]

TAD border deletion at the *Kit* locus causes tissue-specific ectopic activation of a neighboring geneREVIEWER COMMENTS

Reviewer #1 (Remarks to the Author):

Mammalian genomes are partitioned into so-called topologically associated domains (TADs), fundamental genomic units that restrict enhancer-gene interactions. Deleting TAD boundaries typically results in enhancer adoption and erroneous gene activations, indicating the importance of TADs for preventing enhancers and promoters from neighboring TADs from cross-activating each other. How exactly TAD boundaries restrict E-P interactions is not well understood, and in many cases deleting TAD boundaries does not always result in enhancer adoption, indicating complex mechanisms at play. In the present study, Kabirova and colleagues investigated the consequences of TAD boundary deletions in the *Pdgfra/Pdgfra/Kit* genetic locus in three mouse cell types. Each of these genes is differentially expressed in those cell types and *Pdgfra/Kit/Kdr* are located in a separate TAD representing a suitable model to study the effects of TAD boundary deletions. The authors generated several mouse lines in which each of the two TAD boundaries was deleted, followed up by comprehensive gene expression, epigenomic and 3D-genome profiling in three different primary cell lines isolated from mutant mice. The authors demonstrate that both boundaries are required to produce a normal mouse as deletion of each of the TAD boundaries produces strong phenotypes (lethality for *Pdgfra/Kit* boundary and fur coloration phenotype for *Pdgfra/Kit* boundary). Deletion of *Pdgfra/Kit* boundary also knocked out a critical *Pdgfra* gene itself, so it is not possible to separate the cause of the lethal phenotype for *Pdgfra/Kit* boundary deletion. Molecular phenotyping showed that TAD boundary deletion resulted in a fusion of neighboring TADs and enhancer adoption only for *Kit/Kdr* boundary. Interestingly, enhancer adoption was observed only in mast cells, despite the fact that *Kit* gene was active in mast cells and melanocytes.

The study is performed in vivo and in primary cell lines with comprehensive phenotyping, which is a strength of the study. However, a demonstration that TAD boundaries are required for normal gene expression and organismal function is not particularly novel given a large body of work on multiple other mouse genomic loci (e.g., *WNT6/IHH/EPHA4/PAX3*, *HOX*, and many others). A main novel finding is that TAD boundary deletion causes enhancer adoption only in certain cell types in which the hijacking enhancer is active (but not in others). This finding is certainly interesting and is reflected in the title of the paper. However, the authors do not go beyond the observation, and possible mechanisms for such cell-type specificity are not explored (see major comment below), which weakens the study.

Experiments are mostly well executed, although some technical comments need to be addressed. The manuscript is mostly well-written, but it could benefit from editing to improve grammar and readability. Figures could be rearranged to improve the flow and presentation for the reader. Overall, it is a nice study, but conceptual improvements and technical comments must be addressed before publication.

Major Comments

1) Tissue-specific enhancer adoption is a novel finding which is reflected in the title and abstract. However, the authors do not go beyond simple observation. Why does the *Kdr* gene fail to get ectopically activated by *Kit* enhancers after border deletion? Is *Kdr* epigenetically silenced in mast cells but not melanocytes? H3K27me3 ChIP-seq or bisulfite sequencing could, for example, address this question.

2) The authors state that "*Pdgfra/Kit* boundary CBSs deletions were insufficient to change the regulatory and architectural patterns". However, the evidence for this is rather weak. In Fig 3C the read density in *Pdgfra* $\Delta 60K$ is much lower than in WT sample, potentially contributing to the failure to see a sufficient number of interactions to detect TAD fusion. Another possibility is that enhancer adoption will lead to ectopic expression but in other cell types. The data provided is only for MEFs. I understand that it's hard to derive melanocytes and mast cells from homozygous mice, but if there is a gain in expression, it should also be detectable in heterozygous mice.

3) On several occasions, the authors use the H3K27ac mark to assign enhancers to *Pdgfra/*

Pdgfra/Kit. However, no specifics are provided. What is other evidence for their functionality besides H3K27ac? Are they open? H3K27ac is only predictive of enhancer activity and many H3K27ac regions do not act as enhancers. Specific enhancers should be marked in figures.

4) (Related to the previous comment) What is the evidence that these enhancers actually regulate corresponding target genes? Were they deleted in other studies? At a minimum, please provide virtual-4C plots to show physical contacts.

5) All Figures should also show the quantification of expression level for all the genes in the three TADs, since TAD boundary disruption would not only affect Pdgfra, Kit and Kdr but also other genes within the same TAD.

6) It looks like CLOCK also located in the Kdr TAD. Is there an impact on circadian rhythm in the Kit/Kdr TAD disrupted mice?

7) In Fig5G, the explanation of results (lines 313-314 -> 98% of KDR transcripts from the Kit Δ 30K chromosome) is the opposite of what's shown in the figure (98% of KDR transcripts from the Castaneus, orange, chromosome), making it hard to understand the rest of the data in the panel (unsure of were the data got swapped). There is a difference in Pdgfra expression, please comment on that. Please also show the expression data for other genes, is any important pathway related to melanocyte development being disrupted? There's no statistical test on the difference between wildtype and Kit Δ 30k allele.

8) Figs 5D and 5E need some sort of quantification of double-positive cells in deletion vs wild type to confirm the ectopic expression of Kdr in Melanocytes. Also, how do we know that these cells are melanocytes?

Other Comments

1) All figures: the RNA-seq, H3K27ac, and CTCF tracks are too small. The reader can barely see a signal.

2) Perhaps it would be helpful to restructure figure panels to improve clarity.

a. Panels in Fig. 1 can be rearranged to show all data (expression, quantification, contact map) in the same row for each cell type. Please also show expression data for other genes in the same TAD.

b. The placement of Figure 2A is weird. It seems like the authors show the conclusion before the data. Or is this a hypothesis? Then please add a question mark. Otherwise, it should be moved to the end.

c. In Fig 4, please match the image of cells in the same row with the contact maps for each cell type.

d. Please add H3K27ac data for wild-type melanocytes in Fig. 4D, to keep the ordering of the tracks consistent.

3) Fig 3 doesn't specify what the "B" on the figure means, boundary site?

4) Fig. 5C is redundant to 5D

5) Figure sup 12 can be combined with Fig. 2A as final main figure showing the summary of the results

Reviewer #2 (Remarks to the Author):

In the manuscript "TAD border deletion at the Kit locus causes tissue-specific ectopic activation of a neighboring gene", the authors study the impact of TAD 3D architecture disruption on gene regulation. While the results are interesting, they are not novel or unexpected. These results do not significantly impact the field of 3D genome organization and gene regulation. It is true that the authors perform a cell-type-specific study of the effect of 3D chromatin changes on gene expression regulation. They also use an original strategy by breeding the *Mus musculus* with the *Mus castaneus* to demonstrate further their results. However, the study does not add any

additional understanding to the field of 3D chromatin folding or gene regulation. The presented results support more the use of cell-type-specific 3D chromatin data for predicting the consequences of the TAD boundaries removal.

Overall, there are many mistakes reported in the manuscript (see minor comments below). The figures are not always explicit and the structure of the paper does not make it easy to follow and understand. Moreover, some paragraphs from the results part are unnecessary and could be included in other parts of the results.

Major comments:

1- The authors do not mention in their manuscript whether the enhancers they refer to in Fig 1 have already been characterized functionally in the different cell types. Could the authors give more details about this? Have these enhancers been previously annotated functionally or were they just predicted using H3K27ac marks?

2- Is there a difference in the level of CTCF binding within the *Pdgfra* promoter in the Delta60K mutant, explaining the compensation mechanism, such as reported for the SHH locus (Paliou et al., 2019)?

3- The authors state that in the Delta60K mutant, an intergenic region between *Pdgfra* and *Kit* is transcribed, starting from the intact *Pdgfra* promoter and continuing to the nearest termination site (Lines 190-192). First of all, we cannot at all distinguish the presence of this transcript on the Figure. Can the authors, please, provide a supplementary figure showing this. Second, could that also explain the compensation mechanism? The authors can, for example, block the transcription from the *Pdgfra* promoter using a polyA signal to see what would be the effect in this case.

4- Another additional allele that would be important to investigate is a mutation of the remaining CTCF binding site within the *Pdgfra* promoter, along with blocking the transcription from the *Pdgfra* promoter, in the Delta60k mutant.

5- Can the authors please explain the sentence "Consistent with our transcriptome analysis, an H3K27ac-marked enhancer region of the *Pdgfra/Kit* locus remained similar between the mutant and wild-type MEFs lines" (Lines 193-195)? Why do the authors expect any change in H3K27ac that could explain an increase in *Kit* expression? Isn't an enhancer adoption mechanism that is not happening in this case? Which is completely independent from a change in H3K27ac-marked enhancer region.

6- In Supplementary Fig 4, the CTCF and H3K27ac ChIP-seq profiles are completely different between the Wt and the *Kit* Delta30k+ mutant. Can the authors explain this?

7- The paragraphs "Generated mouse strains carrying boundary CBSs deletions" and "Eliminating the contribution of distance decrease to the Hi-C interaction profiles in mutant cells" can be removed and the manuscript structure modified by including the data within these two paragraphs in the other paragraphs. This structure produces redundancy in the text between the paragraphs. It also makes the reader wonder about some results, but those come only later on (just due to the current structure). Talking about the Delta2kb deletion with its full results than the Delta60kb with its full results in comparison to Delta2Kb...etc. would make the manuscript much easier to follow and give a nice structure to the manuscript.

8- The authors stipulate that the sub-structure of the *Kit* TAD into two self-interacting regions in the melanocytes and mast cells can be explained by the high-level transcription of the *Kit* gene. Can the authors block the transcription of the *Kit* gene by introducing a polyA terminating signal, and study the impact on 3D chromatin folding as well the effect on gene expression upon TAD border deletion.

9- Supplementary Fig 11 is a result that should be included in the results part and not shown and discussed in the discussion part.

Minor comments:

- 1- It is not clear in Fig 2 which CTCF sites are disrupted by the 2kb deletion, a magnification showing clearly this is necessary or adding a supplementary figure showing clearly this is necessary.
- 2- Line 147: correct please. With two sites in forward and one in reverse orientation.
- 3- Lines 176-177: correct please. (Supplementary Figure 7)
- 4- Lines 210-211: correct please. Kit with its upstream enhancers are located within the left half of the TAD and thus are insulated from the right half.
- 5- It would be nice to restructure panels c, d, and e from Fig 4 to have the Wt, Kit Delta30k, and the subtraction Hi-C maps in mast cells in one panel, and in parallel the same Hi-C maps in melanocytes. This will make the Figure easy to follow. The same should apply for Fig 3. It would be nice to include the subtraction maps in the main figure to follow easily the results section.
- 6- Line 215: please correct. However, we noted that contacts rewiring mainly involved the left half of the Kit TAD.
- 7- In Fig 4d, there is no H3K27ac for the Wt. However, this is present for the mutant. There is no CTCF data for both the Wt and mutant. Can the authors explain this?
- 8- Line 232: please correct and replace "left" by "right".
- 9- The results presented in Supplementary Fig 7b for the Delta60k mutant contradict the title "Pdgfra/Kit CBSs deletions were insufficient to change the regulatory and architectural patterns", as there is a gain of chromatin contacts in this mutant. However, this is insufficient to drive ectopic gene activation.
- 10- Supplementary Fig 10 is not cited in the text.
- 11- Fig 5c is unnecessary as there is no corresponding Wt with the same magnification.
- 12- The authors should be careful when citing the Figures in the text. It is not possible to cite Fig 4c, d, e before citing Fig 4a, b. This applies to the full manuscript.

Reviewer #3 (Remarks to the Author):

In this manuscript, the authors describe the regulatory landscape and TAD structure at the extended locus of *Pdgfra*, *Kit*, and *Kdr* genes. Comparison of the locus TAD structure in several species revealed the preservation of TAD integrity in tetrapods. They then created multiple genomic deletions in mice that involve CTCF-binding sites at TAD boundaries that separate the regulatory information for these genes. The consequences of CTCF-binding site deletions were analysed in different cellular contexts of cultured cells, reflecting several on/off states (or intermediate states) in gene expression and chromatin state at the locus. Effects on genome folding were assessed by capture-HiC and changes in the transcriptome by RNA-seq. In general, this is an interesting study, which adds to a growing list of *in vivo* studies investigating the consequences of TAD boundary deletions on gene expression. The authors highlight that the choice of the cellular context is important to appreciate the regulatory effects upon CTCF-binding site deletions associated with TAD boundaries. However, I have several concerns which diminish my enthusiasm. Some conclusions appear not to be supported by the data or data are not sufficiently described and presented. Furthermore, the conclusions and their discussion are rather limited. Below I explain my concerns and reservations I have about the manuscript.

Some conclusions seem not to be supported by the data.

First, CTCF ChIP-seq in Melanocytes wild-type and mutant cells are not presented. Lines 343 to 344 state "Notably, the border of an insulated region coincides with a CTCF site at the *Kit* promoter in melanocytes, whereas it shifts toward the *Kit* terminator in mast cells.". However, no CTCF ChIP-seq data are shown or provided by the authors to further assess and compare the described differential CTCF binding at the *Kit* promoter and gene body. This data set is also important to evaluate some of the author's conclusions that internal *Kit* TAD insulation is provided by an epigenetic environment (see lines 241 to 243).

Second, the results from the hybrid experiments, conducted to distinguish between trans and cis

effects, do not align with the manuscript text description and are contradicting. Lines 313 to 314, "We have shown that almost all (98%) Kdr transcripts in hybrid melanocytes originate from the chromosome carrying the Kit Δ 30k deletion (Fig.5g)." While samples from wild-type hybrid cells show equal expression of Kdr from M. mus and M. cas alleles in Figure 5g, the hybrids cells with the mutant allele show exclusive expression from M.cas allele but not M.mus allele that carries the Kit Δ 30k deletion. This contradicts the conclusion in the text. Why is Kdr expression higher from M.cas allele in hybrids with mutant allele compared to Kdr expression from M.cas allele in wild-type hybrids? Why do the authors detect Kdr expression from M.mus allele in wild-type hybrids but not from M.mus allele carrying the Kit Δ 30k deletion? Furthermore, no respective statistical test is shown in Figure 5 g as described in the text, line 315, "The overall level of Kdr expression was significantly lower in the control experiment with M. castaneus x M. musculus Wt hybrids, with both alleles being equally active."

Last, the authors describe differential subTAD structure for Kit TAD in mast cells and melanocytes. Different localization of Kit enhancers might account for intra-TAD insulation (as described in lines 112-114, "According to our H3K27Ac ChIP-seq data (schemes at Fig.1 and Fig.3-4), Pdgfra and Kit enhancers localize upstream of the genes in MEFs and mast cells, respectively. In melanocytes, by contrast, the H3K27ac-rich enhancer region is located downstream."). A major conclusion derived from the manuscript is that the relative position of active enhancers and gene transcription within the subTAD structure determines the insulation defects and gene misexpression in TAD fusion events with Kdr TAD. H3K27ac data for Kit delta30k is shown in Figure 4 but not in wild-type melanocytes. The deposited manuscript data entry shows "melano_h3k27ac_wt.fc.signal.bw" but the data is not provided in the manuscript. In addition, for better appreciation of the schematics in Figures 1 b,c, and d, depicting the "H3K27ac-rich enhancer region", it is highly advisable to cross-reference the depicted schematic enhancer regions in Figure 1 with the actual data figures (e.g. by highlighting enhancer regions in Figure 4).

Addition comments:

In both, mast cells and melanocytes, Kit is highly expressed, but the removal of the TAD border leads to ectopic activation of the Kdr gene in melanocytes only. The authors reasonably discuss and correlate Kdr misexpression with the distribution and accessibility to potential enhancers within the Kit subTADs. In wild-type mast cells Kdr seems not to be expressed and therefore the Kdr promoter might be not permissive to regulatory changes (enhancer adoption) induced by the Kit Δ 30k+ deletion. What is known about Kdr repression (repressive chromatin e.g., H3K27me3) in the investigated cell types or related cell types? Could the repressive chromatin state at the Kdr promoter account for the absence of gene misexpression. Similarly, a repressive chromatin state could account for the absence of gene misexpression of Kit in the Pdgfra Δ 60k mutants. A similar discussion as described above for Kdr could enrich the discussion and conclusions.

The H3K27ac ChIP-seq track in Kit Δ 30k+ mast cells in Supplementary Figure 4 b shows almost a background signal. Could the authors explain the reduced signal compared to the wild type? Is this a technical problem of the ChIP or a consequence of the mutation? This needs to be clarified to fully comprehend the results.

Figure 1a depicts relative gene expression by gene ranking (in %). This representation does not allow a comparison of gene expression between cell types or an understanding of quantitative assessments of gene expression as done by the authors in the text. E.g. lines 202 to 203 "Kdr is normally inactive in mast cells and is marginally expressed in melanocytes (Fig.1a)." or lines 344 to 345 "Though Kit is expressed in both cell types, the expression level in mast cells is 10 times higher." At least a graph or table should be included in the supplementary data, comparing normalized counts (e.g. transcripts per million) of relevant genes at the locus and between tissues. In addition, gene expression levels for all the mutant alleles should be incorporated into such a data table or figure. How do transcriptional changes in Kit Δ 30K and Kit Δ 30K+ compare to each other? Does the increased ectopic signal in the Kit Δ 30K+ deletion allele results in more pronounced Kdr misexpression?

The display of CTCF binding sites in Figure 2 does not match the text description. Lines 132 to 134, "Pdgfra TAD border is characterized by three strong CBSs in reverse orientation, located

within the *Pdgfra* gene body (Fig.2c). The *Pdgfra* Δ 2k strain carries a 2 kb deletion of the three CTCF peaks at the *Pdgfra* TAD border." First, Figure 2 shows three CTCF motifs in reverse orientation in the *Pdgfra* gene body. However, Supplementary Figure 3 and the magnification to a 7 kb region (supposedly comprising the *Pdgfra* promoter) shows CTCF binding and motif orientation of two reverse and one forward-facing CTCF motif and a further distally located CTCF binding site with reverse orientation. Second, a 2 kb deletion does not match the described and displayed genomic size of three CTCF binding sites in reverse orientation. Which CTCF binding sites are deleted in *Pdgfra* Δ 2k allele? Furthermore, lines 140 to 142, "...*Pdgfra* Δ 60k strain carries a 60 kb deletion spanning the whole *Pdgfra*/*Kit* TAD boundary region, and therefore removing the *Pdgfra* coding sequence, leaving only 3 out of 22 exons and a promoter." Can the authors clarify the overlap of *Pdgfra* Δ 60k and *Pdgfra* Δ 2k deletion? Does the *Pdgfra* Δ 60k allele delete promoter proximal CTCF binding sites (Magnification in Suppl. Figure 3) although the deletion allele leaves the promoter region and three exons intact as described in the text?

The authors should revise the manuscript for the description of CTCF motif orientation. For instance, in lines 146 to 147, "The *Kit*/*Kdr* boundary is a joint between two adjacent TADs which contains three clustered CBSs, with two sites in reverse and one in forward orientation (Fig.2c)." According to the definition in the previous manuscript text, Supplementary Figure 3 description, and display of CTCF motif orientation the *Kit*/*Kdr* boundary would consist of one reverse and two forward-oriented CTCF motifs.

Lines 209 to 211, "Contact maps from the wild-type mast cells denoted a nested substructure of the *Kit* TAD, with two self-interacting regions. *Kit* with its upstream enhancers are located within the right half of the TAD and thus are insulated from the left half (Fig.1c)." and 216 to 217 "The insulation inside the *Kit* TAD was not affected by the deletion, thus, a self-interacting region, located at the right side, remained isolated." The description of "right half" or "right side" does not match schematic in Figure 1c for mast cells. According to Figure 1 c, enhancers in mast cells are located in the left half of *Kit* TAD. Is this the case? See also line 232 and 235, "left half" instead of right half? In general, the description of genomic orientation and position is not consistent in the manuscript, using terms such as right, left, upstream, or downstream of different genes. The readability of the manuscript should be improved by using consistent formatting for relative positions. For instance, referring to relative positions by using the centromeric and telomeric orientation of the chromosome (or similar).

Line 190 to 192, "In *Pdgfra* Δ 60k, an intergenic region between *Pdgfra* and *Kit* is transcribed, starting from the intact *Pdgfra* promoter and continuing to the nearest termination site." Could the authors highlight and/or enlarge the region in the respective figures? It is not clear to the reader what the authors are referring to.

In the discussion the authors describe a larger 300kb deletion, involving the *Kit*/*Kdr* border, which has no gene expression consequences on *Kdr*. However, the description lacks a deeper relation to the produced data and conclusions. According to the model presented in Supplementary Figure 12 (Mast cells), the 300 kb deletion removes the internal *Kit* TAD border and therefore brings *Kdr* in close proximity to potential mast cell active enhancers. Could the authors describe why they think that the direct proximity of *Kdr* to these enhancers has no consequences? For instance, to what extent is the *Kdr* promoter permissive to the regulatory input (of *Kit* landscape) in mast cells?

Line 289 says "However, the analysis did not reveal significant differences between two genotypes;" and the description of Supplementary Figure 10 refers to the quantification of *Kit*-positive cells in melanocytes. No quantification is shown in the manuscript. In the current state Supplementary Figure 10 does not provide information to the reader.

The terms like "conservativity" and "conservatism" or "since TADs are conservative" (line 435) are not appropriate word choices to describe conserved relative TAD position and integrity between species.

In lines 176 to 177. Wrong figure reference. Supplementary Fig.4 and Fig.5a,b both refer to *Kit* Δ 30+ allele but not *Pdgfra* Δ alleles. A likely reference is Supplementary Figure 7?

Wrong Figure reference in line 220. Are the authors referring instead to Supplementary Figure 7?

Wrong Figure reference in line 280. Are the authors referring instead to Supplementary Figure 8?

REVIEWER COMMENTS

Reviewer #1 (Remarks to the Author):

Mammalian genomes are partitioned into so-called topologically associated domains (TADs), fundamental genomic units that restrict enhancer-gene interactions. Deleting TAD boundaries typically results in enhancer adoption and erroneous gene activations, indicating the importance of TADs for preventing enhancers and promoters from neighboring TADs from cross-activating each other. How exactly TAD boundaries restrict E-P interactions is not well understood, and in many cases deleting TAD boundaries does not always result in enhancer adoption, indicating complex mechanisms at play. In the present study, Kabirova and colleagues investigated the consequences of TAD boundary deletions in the *Pdgfra/Pdgfra/Kit* genetic locus in three mouse cell types. Each of these genes is differentially expressed in those cell types and *Pdgfra/Kit/Kdr* are located in a separate TAD representing a suitable model to study the effects of TAD boundary deletions. The authors generated several mouse lines in which each of the two TAD boundaries was deleted, followed up by comprehensive gene expression, epigenomic and 3D-genome profiling in three different primary cell lines isolated from mutant mice. The authors demonstrate that both boundaries are required to produce a normal mouse as deletion of each of the TAD boundaries produces strong phenotypes (lethality for *Pdgfra/Kit* boundary and fur coloration phenotype for *Pdgfra/Kit* boundary). Deletion of *Pdgfra/Kit* boundary also knocked out a critical *Pdgfra* gene itself, so it is not possible to separate the cause of the lethal phenotype for *Pdgfra/Kit* boundary deletion. Molecular phenotyping showed that TAD boundary deletion resulted in a fusion of neighboring TADs and enhancer adoption only for *Kit/Kdr* boundary. Interestingly, enhancer adoption was observed only in mast cells, despite the fact that *Kit* gene was active in mast cells and melanocytes.

The study is performed *in vivo* and in primary cell lines with comprehensive phenotyping, which is a strength of the study. However, a demonstration that TAD boundaries are required for normal gene expression and organismal function is not particularly novel given a large body of work on multiple other mouse genomic loci (e.g., *WNT6/IHH/EPHA4/PAX3*, *HOX*, and many others). A main novel finding is that TAD boundary deletion causes enhancer adoption only in certain cell types in which the hijacking enhancer is active (but not in others). This finding is certainly interesting and is reflected in the title of the paper. However, the authors do not go beyond the observation, and possible mechanisms for such cell-type specificity are not explored (see major comment below), which weakens the study.

Experiments are mostly well executed, although some technical comments need to be addressed. The manuscript is mostly well-written, but it could benefit from

editing to improve grammar and readability. Figures could be rearranged to improve the flow and presentation for the reader. Overall, it is a nice study, but conceptual improvements and technical comments must be addressed before publication.

Major comments

1) Tissue-specific enhancer adoption is a novel finding which is reflected in the title and abstract. However, the authors do not go beyond simple observation. Why does the *Kdr* gene fail to get ectopically activated by *Kit* enhancers after border deletion? Is *Kdr* epigenetically silenced in mast cells but not melanocytes? H3K27me3 ChIP-seq or bisulfite sequencing could, for example, address this question.

Thank you for your comment. The promoter's epigenetic state can indeed affect the transcriptional output of an enhancer hijacking. In Botten et al.¹ the effect of a t(5;14) translocation was studied. They have shown that t(5;14) causes *TLX3* proto-oncogene activation through hijacking of active *BCL11B* enhancers in several leukemic cell lines. Strikingly, engineering of t(5;14) in Jurkat cells induced de novo enhancer-promoter looping without *TLX3* activation. The lack of *TLX3* activation in Jurkat cells despite the formation of ectopic loops was associated with hypermethylation of the *TLX3* promoter. *TLX3*-expressing cell lines displayed lower methylation levels, allowing for the aberrant activation.

Another example of how DNA methylation can control the enhancer usage is independent gene regulation in a multi-gene TAD². Methylation-induced promoter silencing of the *Zfp42R* gene serves as one of tissue-specific mechanisms allowing its differential expression. Despite contacting active *Fat1* enhancers in a shared TAD in limb tissue, methylated *Zfp42* promoter remains irresponsible for activation.

However, in two cases mentioned above, enhancer-promoter contacts between two genes are observed, as evidenced from 4C or Hi-C data, leading to gene activation in the absence of DNA methylation. While in our study *Kit* enhancers remain separated from the *Kdr* promoter in mast cells, which probably defined the transcriptional output of the deletion in this cell type.

2) The authors state that “*Pdgfra/Kit* boundary CBSs deletions were insufficient to change the regulatory and architectural patterns”. However, the evidence for this is rather weak. In Fig 3C the read density in *Pdgfra* Δ 60K is much lower than in WT sample, potentially contributing to the failure to see a sufficient number of interactions to detect TAD fusion.

Another possibility is that enhancer adoption will lead to ectopic expression but in other cell types. The data provided is only for MEFs. I understand that it's hard to derive melanocytes and mast cells from homozygous mice, but if there is a gain in expression, it should also be detectable in heterozygous mice.

Thank you for your comment. We agree that further elaborating on the effects of the *Pdgfra* Δ 60K deletion using new data — that is, providing a higher density Hi-C map and expanding the cell types tested — would be helpful. However, we believe that expanding our dataset is neither feasible, given the costs involved, nor would significantly support our argument.

Though higher read density would allow us to better see spatial contacts, we believe that the improvements would affect only minor changes in spatial contact or increase already visible strong contacts. Taking into account that a remaining CTCF motif at the *Pdgfra* side of the *Pdgfra/Kit* border possibly played a compensatory role, we argue that the TADs remained largely separate. Subtraction maps (see modified Fig.3e,c) do show a gain of contact in the inter-TAD region but it is not sufficient for an ectopic activation or TADs fusion. If it was not the case, and a new 3D structure was established in *Pdgfra* Δ 60K, we believe contact enrichment between *Pdgfra* and *Kit* genes would be visible on the map we provide, as there are other stable 3D structures visible, such as TADs.

It is a good point that an enhancer adoption could lead to an ectopic gene activation in the cell types that we have not tested. In addition to fibroblasts, other cell types with a suitable expression pattern (*Pdgfra* is active, *Kit* is silent) could be: other mesenchymal cells, leydig cells, oligodendrocyte precursor cells, smooth muscle cells. However, we believe that we would encounter the same effect in these cell types due to *Pdgfra* localization at a TAD border, which provides robust insulation.

As to melanocytes and mast cells, *Kit* is already highly active in them, while *Pdgfra* is somewhat expressed in melanocytes and silent in mast cells. So a reverse experiment (*Kit* is active, *Pdgfra* is silent) could be conducted, where we would expect to detect an ectopic activation of *Pdgfra*. However, the *Pdgfra/Kit* border deletion requires knockout of *Pdgfra*, so there would be no gene to test the activation of.

Overall, our model of *Pdgfra/Kit* TAD border shows how robust the regulation is for such a crucial development gene as *Pdgfra*. It is known that house-keeping genes are enriched near topological boundary regions³ and transcription itself has been shown to act as a barrier for cohesin-mediated loop extrusion⁴. Our

data coincides with that suggesting that *Pdgfra* location at a TAD boundary provides advantages for robust gene regulation.

Your notion inspired us to shift our narration from insufficiency of the deletions to what it means. To highlight that gene located at a TAD border could strengthen the border we changed the name of the chapter from “*Pdgfra/Kit boundary CBSs deletions were insufficient to change the regulatory and architectural patterns*” to “*Pdgfra gene localisation at the TAD border challenges the border disruption*”. We have revised the text to address your concerns and hope that it is now clearer.

3) On several occasions, the authors use the H3K27ac mark to assign enhancers to *Pdgfra/Kit*. However, no specifics are provided. What is other evidence for their functionality besides H3K27ac? Are they open? H3K27ac is only predictive of enhancer activity and many h3k27ac regions do not act as enhancers. Specific enhancers should be marked in figures.

Thank you for pointing this out. We do agree that assignment of enhancers based on H3K27ac enrichment solely is non-exhaustive. To provide clarity we explicitly described it in the manuscript.

“Here we refer to enhancers based on enrichment of an enhancer-related epigenetic mark H3K27ac.”

We updated Fig.1 to highlight the enhancer regions and to emphasize the limitation we introduced by referring to enhancers as H3K27ac regions.

To better characterise the regulation context of the *Kit* locus we expanded our H3K27ac data with the data found by other studies on enrichment of enhancer-(H3K4me1) and heterochromatin-associated (H3K27me3) epigenetic marks, binding of transcription factors (Gata2, Fos, JunD, p300, CEBP), and chromatin accessibility (ATAC-seq). We introduced a figure with the data as a Supplementary Figure 2.

For mast cells, ATAC-seq demonstrates that open chromatin is mostly located upstream to *Kit* and it coincides with H3K27me3, which spans all the locus downstream from *Kit* gene. For melanocytes, ATAC-seq demonstrates open chromatin surrounding *Kit* from both sides. For MEFs, ATAC-seq confirms open chromatin located upstream to *Pdgfra* overlapping signals from our H3K27ac data. Additionally, AP-1 transcription factors Fos and JunD, required for enhancer selection in fibroblasts⁵, are revealed to bind upstream to *Pdgfra*. Thus,

expanding our data with other studies allows us to ensure that we are able to refer to these regions as enhancers, though with a limitation that we are only able to refer to them as an enhancer upstream or downstream to a gene rather than an exact region.

We intended to functionally characterise the enhancer regions at the *Kit* locus using a luciferase reporter assay. We chose several enhancers based on an expanded characterisation of the regulatory context in the locus (“possible enhancers” in Fig.1 here). We used sequences of the chosen regions to clone them into a pGL4.23 vector containing the luciferase reporter gene under a minimal promoter. Comparing luciferase expression from the vectors containing enhancer sequences with vectors lacking them would allow us to dissect whether the chosen sequences provide activation of a gene. To conduct the experiment we looked for cell cultures that would represent mast cells and melanocytes, could be transfected easily and passaged over a longer period than primary cell cultures. We found suitable cell cultures to be a mouse mastocytoma cell line P815 and a mouse melanoma cell line B16F10, and we also had access to a human melanoma cell line Sk-mel-2. Mainly we expected the luciferase reporter assay to reveal that enhancer regions downstream to *Kit* are active only in melanocytes. However we did not detect a significant difference between enhancer and control vectors for any chosen enhancer neither in melanocytes (B16F10, Sk-mel-2), nor in mast cells (P815). We performed the same experiment in HEK293T cells, as a commonly used cell type to functionally test enhancers, but there was no difference between control and enhancer vectors.

There are several possible explanations for the lack of increase in luciferase expression by the chosen enhancers. It could be explained by insufficiency of transfection in the chosen cell lines. Another possible explanation could be a need for tissue-specific factors to promote enhancer activation that may be present in primary cell cultures and not in tumor cells, moreover not in the non-target cell line HEK293T. Since at least one region we have tested was confirmed to function as an enhancer through a series of deletions by another study⁶, we considered the lack of activation in our experiment to be a technical problem. Considering the luciferase reporter assay was not informative, we decided to not include it in the manuscript.

We acknowledge a lack of a functional test of enhancers is a limitation of the study. However we believe that by expanding regulatory characterisation of the locus by other studies we provide more certainty to the enhancers assignment.

Fig.1 Regulatory context of the *Kit* locus. Characteristics of regions around *Kit* and *Kdr* (a), and *Pdgfra* (b) genes. H3K27ac data is from our study. Other data taken from Li Y. et al.⁷, Calero-Nieto F. J. et al.⁸, Berrozpe G. et al.⁹, Infarinato N. R. et al.¹⁰, Vierbuchen T. et al.⁵, Zhang S. et al.¹¹, Phan Q. M. et al.¹².

4) (Related to the previous comment) What is the evidence that these enhancers actually regulate corresponding target genes? Were they deleted in other studies? At a minimum, please provide virtual-4C plots to show physical contacts.

Thank you for your comment. There is indeed evidence of correspondence between the *Kit* gene and its enhancers. An experiment deleting the enhancers located upstream to *Kit* was conducted by Berrozpe G. and colleagues⁶. They have identified two enhancer regions in bone marrow mast cells and melanocytes located at 147 to 154 kb and 21 to 28 kb upstream of the *Kit* transcription start site, respectively. They revealed that distant upstream regions are required for *Kit* expression in mast cells (upon their deletion *Kit* was not

transcribed), but not in melanocytes, for a more proximal regulatory region is essential in melanocytes. Interestingly, it coincides with our Hi-C maps, where the Kit TAD is heterogeneous and Kit is somewhat insulated from the upstream “mast cells” enhancers in melanocytes.

Accounting for the given suggestion, we generated virtual 4C tracks. We have used the ‘generate 1D track’ function of Juicebox on our Hi-C maps (resolution 1 kb) from wild-type and mutant cells. For viewpoints we chose promoters of the genes *Pdgfra*, *Kit*, *Kdr* and enhancers based on enrichment of enhancer- and heterochromatin-associated epigenetic marks, binding of transcription factors, and chromatin accessibility (Fig.1 here). We provide here genome coordinates of the chosen enhancers (Table 1 here). Of note, an enhancer MC1 overlaps the enhancer region identified by Berrozpe G. and colleagues.

Table 1. Genome coordinates for enhancers chosen as viewpoints for a virtual 4C.

Name	Genome coordinates, mm10
MEF1	chr5:74752806-74753434
MEF2	chr5:74777292-74777900
MEF3	chr5:74897034-74897744
MEF4	chr5:74974177-74975365
MEF5	chr5:75047016-75047438
MC1	chr5:75429273-75429997
MC2	chr5:75432841-75433510
MC3	chr5:75460325-75461040
MC4	chr5:75462108-75463053
MC_poised1	chr5:75804289-75804888
MC_poised2	chr5:75826375-75826874
MelanoUp1	chr5:75346780-75347244
MelanoUp2	chr5:75355301-75356059
MelanoUp3	chr5:75454137-75454800
MelanoUp4	chr5:75465595-75466105

MelanoDown1	chr5:75657589-75658168
MelanoDown2	chr5:75661917-75662554
MelanoDown3	chr5:75683819-75684849

The obtained virtual 4C tracks (Fig.2 here) of the chosen enhancers demonstrate higher peaks near the corresponding gene promoters (either *Pdgfra* for MEFs or *Kit* for the rest). For tracks of a promoter as a viewpoint several higher peaks correspond to H3K27ac data. Also melanocytes-associated enhancers in mast cells and mast cells associated enhancers in melanocytes demonstrate lower peaks in *Kit* promoter, thus suggesting a plausible tissue specificity.

Interestingly, in case of *Kdr* ectopic activation in melanocytes, enhancers located downstream to *Kit* (MelanoDown 2, 3) demonstrate enrichment of contacts with *Kdr* collocating with the whole gene. While other enhancers (MelanoUp) demonstrate an appearance of new contacts with *Kdr*, but not necessarily higher than other regions.

However, the difference of the described peaks from the background is marginal in most cases. It is consistent with the fact that enhancer-promoter dots are not well distinguished in Hi-C maps underlying virtual 4C analysis. It can be described by the fact that CTCF-mediated loops are generally stronger than enhancer-mediated loops¹³.

Thus, we are unable to pinpoint exactly targets for the enhancers in the *Kit* locus. But considering the fact that (1) upstream enhancers were shown to be crucial for *Kit* activation in mast cells and not in melanocytes, and (2) virtual 4C demonstrated a well visible enrichment of contacts between enhancers downstream to *Kit* with *Kdr* in melanocytes, we believe hijacking of downstream *Kit* enhancers is a plausible explanation for *Kdr* activation in melanocytes.

a

ChIP-seq H3K27Ac Wt

Pdgfra: Wt

Pdgfra: 60k del

MEF1: Wt

MEF1: 60k del

MEF2: Wt

MEF2: 60k del

MEF3: Wt

MEF3: 60k del

MEF4: Wt

MEF4: 60k del

MEF5: Wt

MEF5: 60k del

Kit: Wt

Kit: 60k del

**b**

ChIP-seq H3K27Ac Wt

mast cells

MC1: Wt

MC1: kit30k del

MC2: Wt

MC2: kit30k del

MC3: Wt

MC3: kit30k del

MC4: Wt

MC4: kit30k del

Kit: Wt

Kit: kit30k del

MC poised1: Wt

MC poised1: kit30k del

MC poised2: Wt

MC poised2: kit30k del

Fig.2. Virtual 4C. Enhancers and target genes chosen as viewpoints for MEFs (a), mast cells (b) and melanocytes (c) are marked by an eye symbol. Gray squares and arrows point to the most noticeable difference in spatial contacts.

5) All Figures should also show the quantification of expression level for all the genes in the three TADs, since TAD boundary disruption would not only affect *Pdgfra*, *Kit* and *Kdr* but also other genes within the same TAD.

Thank you for your comment. Due to the hierarchical structure of TADs, including nested TADs, it is challenging to precisely define the number of genes within a specific TAD. The *Kit* gene is definitively the sole gene in its TAD. However, for the *Pdgfra* and *Kdr* genes, determining the boundaries of their respective TADs is more complex, making it difficult to ascertain the complete list of genes within these TADs. To avoid confusion regarding this issue, we have explicitly stated in the text that we examined all genes within a 2-megabase range from the deletion sites. Our findings indicate that, except for *Kdr*, no other gene within this range exhibited significant changes in expression.

6) It looks like *CLOCK* also located in the *Kdr* TAD. Is there an impact on circadian rhythm in the *Kit/Kdr* TAD disrupted mice?

Thank you for your suggestion. We acknowledge the proximity of the key gene regulating circadian rhythm near the *Kit* locus. Initially, since there was no change in the expression of the *Clock* gene, we did not investigate changes in circadian rhythm in our original article. However, your attention to this gene, coupled with the possibility that *Clock* gene expression might vary in other cell types due to the removal of the *Kit/Kdr* TAD boundary, prompted further investigation. Consequently, we conducted experiments assessing locomotor activity, sleep, and food and water intake in *Kit30* and wild-type mice. These additional data have been included in the revised manuscript. Our findings indicate no significant differences in these parameters between the groups, suggesting that the *Kit30* deletion does not impact the regulation of the *Clock* gene.

7) In Fig5G, the explanation of results (lines 313-314 -> 98% of KDR transcripts from the *Kit* Δ 30K chromosome) is the opposite of what's shown in the figure (98% of KDR transcripts from the *Castaneus*, orange, chromosome), making it hard to understand the rest of the data in the panel (unsure of were the data got swapped). There is a difference in *Pdgfra* expression, please comment on that. Please also show the expression data for other genes, is any important pathway related to

melanocyte development being disrupted? There's no statistical test on the difference between wildtype and Kit Δ30k allele.

Thank you for noting that the alleles caption on the figure got swapped. The information in the text description (98% of the Kdr transcripts in hybrid melanocytes originate from the chromosome carrying the Kit Δ30k deletion) is what we detect from our RNA-seq data analysis and what we intended to illustrate by the figure.

The absence of statistical test on the difference between wildtype and Kit Δ30k allele was indeed an omission, which was also mentioned by Reviewer 3 in major comment 2. Inspired by your suggestion we have changed our computational approach to better illustrate the difference between wild-type and KitΔ30k allele. Instead of counting the proportion of reads carrying SNPs from one of the two species, we have compared normalized read counts from wild-type and KitΔ30k, using read counts from the *M. castaneus* allele as a normalization factor. The significance of differences was confirmed by the Mann-Whitney test. This analysis also takes into account the difference in *Pdgfra* expression, which turned out to be insignificant. Please see Fig.5e.

“We isolated primary cells from the epidermis of the hybrids, differentiated them, performed RNA-seq in melanocytes, and calculated the read counts from one of the two species based on the SNP analyses. Then, we used read counts from the M. castaneus allele as a normalization factor and compared the level of Pdgfra, Kit and Kdr expression between wild-type and Kit Δ30k alleles.”

Identification of possible melanocyte related pathways using KitΔ30k RNA-seq data would be an essential improvement for our study. We have performed GO analysis using the list of differentially expressed genes and found in total 52 terms that were enriched. Among them, the terms GO:004247 (melanosome) and GO:004877 (pigment granule) suggest some relationships to melanocyte development. However, the biological interpretation of this data is intricate. Some of the key genes involved in melanosome maturation and melanin synthesis, such as *Pmel*, *Tyr* and *Tyrp1* are significantly upregulated in KitΔ30k melanocytes, which was an unexpected result, given their paler phenotype. Most likely, a comprehensive analysis of the secondary effects of the changes we introduced to the genome structure requires a more complex approach. Since the main focus of our study was to identify the direct effects of boundary deletion on ectopic activation of neighboring genes, further interpretation of the mechanisms is a matter of future studies.

8) Figs 5D and 5E need some sort of quantification of double-positive cells in deletion vs wild type to confirm the ectopic expression of Kdr in Melanocytes. Also, how do we know that these cells are melanocytes?

Thank you for this fair notion. We have replaced the immunostained sections of the vibrissae region with more representative images of the skin sections from the embryo's back. Multiple Kit-positive cells can be seen on the images displaying distinct dendritic morphology (please see Supplementary Fig X). In Kit Δ 30k essentially all these cells demonstrate double-positive staining, which makes the quantitative assessment a complicated task. The exact identity of these cells is indeed undetermined, as we have not used melanocyte-specific markers to stain our sections. Although, given their location in the epidermal layer by E15.5, characteristic branched morphology and Kit-positive signal, we have good reason to believe these cells are melanocytes¹⁴. We have also rephrased the description of these cells in the text and discussed their possible identity.

“Numerous double-positive cells were found in the skin of Kit Δ 30k embryos (Supplementary Fig.8). These cells were located at the epidermal layer and displayed branched, dendritic morphology. Kit-positive cells with identical shape in the skin of Wt embryos were negative for Kdr signal. Kit-expressing cell types, residing in the mouse fetal skin, are melanocytes and mast cells, although the latter exclusively locate to the dermis. Thus, the location of Kit-Kdr double-positive cells in the epidermal layer and their dendritic morphology suggests that these are melanocytes”

Other Comments

1) All figures: the RNA-seq, H3K27ac, and CTCF tracks are too small. The reader can barely see a signal.

Thank you for the notion. We updated all the figures containing RNA-seq, H3K27ac, and CTCF to better visualize the tracks either by making a picture itself larger, or by zooming in to the discussed genome region.

2) Perhaps it would be helpful to restructure figure panels to improve clarity.
a) Panels in Fig. 1 can be rearranged to show all data (expression, quantification, contact map) in the same row for each cell type. Please also show expression data for other genes in the same TAD.

Thank you for your comment. We updated Fig.1a so that gene expression in all the cell types (mast cells, melanocytes, and MEFs) are visualized on

the same plot. We hope it provides more visual clarity. As a consequence we were able to fit more information on the Fig.1, so we added a figure of research design.

As to expression data for other genes in the same TAD, we are concerned an addition of all other genes of the locus would complicate the figure and confuse a reader. Since we found that within a 2-megabase range from the deletion sites no other gene, except *Kdr* in melanocytes, exhibited significant changes in expression, so we decided to leave other genes out of the figures and discuss them in the text.

b) The placement of Figure 2A is weird. It seems like the authors show the conclusion before the data. Or is this a hypothesis? Then please add a question mark. Otherwise, it should be moved to the end.

Thank you for the comment. It was in fact a hypothesis in Fig.2a. According to your suggestion we did consider adding a question mark, but we found it is more clear to remove an expected result from the figure and make the rest of the picture a part of an experiment design instead. Please see Fig.1b for a modified version. We hope it clarifies the confusion.

c) In Fig 4, please match the image of cells in the same row with the contact maps for each cell type.

Thank you for your suggestion. We updated Fig.4 accordingly.

d) Please add H3K27ac data for wild-type melanocytes in Fig. 4D, to keep the ordering of the tracks consistent.

Thank you for noting, it is in fact a crucial part of the figure. We included the H3K27ac data for Wt melanocytes in the modified Fig.4.

3) Fig 3 doesn't specify what the "B" on the figure means, boundary site?

Thank you for your comment. We did mean a CTCF boundary site by the "B" symbol. However in the modified manuscript we reconsidered usage of the symbol and decided that it does not provide clarity as we intended so we removed it from all the figures. To visually introduce TADs borders, which we refer to, we modified Fig.2 to emphasize where the borders are at the Hi-C maps. We hope it provides a better understanding for a reader.

4) Fig. 5C is redundant to 5D

Thank you for noting this, we have removed the images of the vibrissae region and added more representative images of the skin sections which can be found in the Supplementary section (Supplementary Fig.8).

5) Figure sup 12 can be combined with Fig. 2A as final main figure showing the summary of the results

Thank you for the suggestion. We provided a graphic summary of the results combining these two figures. Please see Fig.6.

Reviewer #2 (Remarks to the Author):

In the manuscript "TAD border deletion at the Kit locus causes tissue-specific ectopic activation of a neighboring gene", the authors study the impact of TAD 3D architecture disruption on gene regulation. While the results are interesting, they are not novel or unexpected. These results do not significantly impact the field of 3D genome organization and gene regulation. It is true that the authors perform a cell-type-specific study of the effect of 3D chromatin changes on gene expression regulation. They also use an original strategy by breeding the *Mus musculus* with the *Mus castaneus* to demonstrate further their results. However, the study does not add any additional understanding to the field of 3D chromatin folding or gene regulation. The presented results support more the use of cell-type-specific 3D chromatin data for predicting the consequences of the TAD boundaries removal. Overall, there are many mistakes reported in the manuscript (see minor comments below). The figures are not always explicit and the structure of the paper does not make it easy to follow and understand. Moreover, some paragraphs from the results part are unnecessary and could be included in other parts of the results.

Major comments:

- 1) The authors do not mention in their manuscript whether the enhancers they refer to in Fig 1 have already been characterized functionally in the different cell types. Could the authors give more details about this? Have these enhancers been previously annotated functionally or were they just predicted using H3K27ac marks?**

Thank you for pointing this out. It is an important question that was also raised by Reviewer 1 in major comments 3 and 4. Here we cite our answer:

(R1 major comment 4)

There is indeed evidence of correspondence between the *Kit* gene and its enhancers. An experiment deleting the enhancers located upstream to *Kit* was conducted by Berrozpe G. and colleagues⁶. They have identified two enhancer regions in bone marrow mast cells and melanocytes located at 147 to 154 kb and 21 to 28 kb upstream of the *Kit* transcription start site, respectively. They revealed that distant upstream regions are required for *Kit* expression in mast cells (upon their deletion *Kit* was not transcribed), but not in melanocytes, for a more proximal regulatory region is essential in melanocytes. Interestingly, it coincides with our Hi-C maps, where the *Kit* TAD is heterogeneous and *Kit* is somewhat insulated from the upstream “mast cells” enhancers in melanocytes.

(R1 major comment 3)

We do agree that assignment of enhancers based on H3K27ac enrichment solely is non-exhaustive. To provide clarity we explicitly described it in the manuscript.

“Here we refer to enhancers based on enrichment of enhancer-related epigenetic mark H3K27ac, which we acknowledge to be a non-exhaustive definition of an enhancer.”

We updated Fig.1 to highlight the enhancer regions and to emphasize the limitation we introduced by referring to enhancers as H3K27ac regions.

To better characterise the regulation context of the *Kit* locus we expanded our H3K27ac data with the data found by other studies on enrichment of enhancer- (H3K4me1) and heterochromatin-associated (H3K27me3) epigenetic marks, binding of transcription factors (*Gata2*, *Fos*, *JunD*, p300, CEBP), and chromatin accessibility (ATAC-seq). We introduced a figure with the data as a Supplementary Figure 2.

For mast cells, ATAC-seq demonstrates that open chromatin is mostly located upstream to *Kit* and it coincides with H3K27me3, which spans all the locus downstream from *Kit* gene. For melanocytes, ATAC-seq demonstrates open chromatin surrounding *Kit* from both sides. For MEFs, ATAC-seq confirms open chromatin located upstream to *Pdgfra* overlapping signals from our H3K27ac data. Additionally, AP-1 transcription factors *Fos* and *JunD*, required for enhancer selection in fibroblasts (ref), are revealed to bind upstream to *Pdgfra*. Thus, expanding

our data with other studies allows us to ensure that we are able to refer to these regions as enhancers, though with a limitation that we are only able to refer to them as an enhancer upstream or downstream to a gene rather than an exact region.

We intended to functionally characterise the enhancer regions at the *Kit* locus using a luciferase reporter assay. We chose several enhancers based on an expanded characterisation of the regulatory context in the locus (“possible enhancers” in Fig.1 here). We used sequences of the chosen regions to clone them into a pGL4.23 vector containing the luciferase reporter gene under a minimal promoter. Comparing luciferase expression from the vectors containing enhancer sequences with vectors lacking them would allow us to dissect whether the chosen sequences provide activation of a gene. To conduct the experiment we looked for cell cultures that would represent mast cells and melanocytes, could be transfected easily and passaged over a longer period than primary cell cultures. We found suitable cell cultures to be a mouse mastocytoma cell line P815 and a mouse melanoma cell line B16F10, and we also had access to a human melanoma cell line Sk-mel-2. Mainly we expected the luciferase reporter assay to reveal that enhancer regions downstream to *Kit* are active only in melanocytes. However we did not detect a significant difference between enhancer and control vectors for any chosen enhancer neither in melanocytes, nor in mast cells.

Since at least one region we have tested was confirmed to function as an enhancer through a series of deletions by another study⁶, we considered the lack of activation in our experiment to be a technical problem. One possible explanation could be a need for tissue-specific factors to promote enhancer activation that may be present in primary cell cultures and not in tumor cells.

We acknowledge a lack of a functional test of enhancers is a limitation of the study. However we believe that by expanding regulatory characterisation of the locus by other studies we provide more certainty to the enhancers assignment.

2) Is there a difference in the level of CTCF binding within the *Pdgfra* promoter in the Delta60K mutant, explaining the compensation mechanism, such as reported for the SHH locus (Paliou et al., 2019)?

Thank you for the constructive comment. In the work by Paliou and colleagues¹⁵ deletions of CTCF sites resulted in the increased CTCF binding at the neighboring ectopic sites, which is evident from their ChIP-seq tracks. In our ChIP data we do see increased CTCF signal at the promoter peak (please see zoom in on the *Pdgfra* promoter region on the Supplementary Fig.4). However, we also can see some differences in the CTCF peaks height on distant CTCF sites, which can not be related to the generated mutations. At the same time, it should be noted that our ChIP-seq replicates reproducibility was confirmed by the IDR analysis, thus, some inconsistency in peak height between experiments might be a feature of a ChIP method. To address this issue we have added the differential peak analysis to our manuscript. Please see Supplementary Fig.4.

*“We have performed differential peak analysis to computationally assess the significance of our observations (Supplementary Fig.4). We have used Homer software which utilizes all the peaks detected across the genome to calculate the differential enrichment between Wt and mutant datasets. The *Pdgfra* promoter CTCF peak appeared to be differentially enriched with the p -value < 0.05 .”*

- 3) The authors state that in the Delta60K mutant, an intergenic region between *Pdgfra* and *Kit* is transcribed, starting from the intact *Pdgfra* promoter and continuing to the nearest termination site (Lines 190-192). First of all, we cannot at all distinguish the presence of this transcript on the Figure. Can the authors, please, provide a supplementary figure showing this. Second, could that also explain the compensation mechanism? The authors can, for example, block the transcription from the *Pdgfra* promoter using a polyA signal to see what would be the effect in this case.**

Thank you for noticing. We have included the expanded view of the intergenic transcript to the Supplementary section. Please see Supplementary Figure №5 .

We agree with the Reviewer that blocking the transcription from the *Pdgfra* promoter in Delta60K would provide useful information for understanding a compensation mechanism upon boundary deletion. However, that would be a serious complication to the study, as it requires either obtaining a mouse strain with an integration of a polyA signal, or incorporating a signal into primary fibroblasts from Delta60K animals. In the first case, we need to generate a 60K+/- PolyA+/ins strain (as blocking the *Pdgfra* transcription would be lethal by E15), and then obtain homozygous embryos. In the second case we face difficulties related to the primary cells transfection, e.g. limited passages, high mortality.

However, we can speculate about the effects of blocking the transcription from the *Pdgfra* promoter. Generally, transcribed boundaries do have a stronger CTCF enrichment and are better insulated than non-transcribed, which is facilitated by CTCF-RNA interactions. Based on the study by Islam et al. 2023¹⁶, this effect is correlated with the transcription level, with highly transcribed boundaries being the most CTCF-enriched. However, despite the positive correlation between boundary RNA levels and insulation, even the least transcribed boundaries are better insulated than nontranscribed. Thus, transcription from the *Pdgfra* promoter in the Delta60K could potentially impact the boundary insulation by reinforcing the CTCF binding in the remaining site within the *Pdgfra* promoter. But, since the level of intergenic mRNA in the Delta60K mutant is low, we would suspect the impact of this transcript on the CTCF binding to be rather weak.

4) Another additional allele that would be important to investigate is a mutation of the remaining CTCF binding site within the *Pdgfra* promoter, along with blocking the transcription from the *Pdgfra* promoter, in the Delta60k mutant.

Thank you for your comment. We agree that it would be of interest to analyze the effects of the *Pdgfra* promoter CTCF deletion on 3D and regulation. But, in addition to the technical difficulties, described in the previous answer, deletion of this remaining CTCF would mean the deletion of the entire *Pdgfra* promoter. Loss of the target promoter was shown to retarget enhancer activity to the non-target genes located in the same TAD¹⁷. It therefore would be difficult to distinguish the effects of boundary deletion from the chromatin rewiring induced by the promoter loss. Disruption of the promoter CTCF motif could be the option, but in the same study by Oh et al. 2021 it reproduced the effects of the entire promoter deletion. For this reason, we believe that deleting the promoter CTCF peak would trigger more extensive perturbations than just boundary deletion.

5) Can the authors please explain the sentence "Consistent with our transcriptome analysis, an H3K27ac-marked enhancer region of the *Pdgfra/Kit* locus remained similar between the mutant and wild-type MEFs lines" (Lines 193-195)? Why do the authors expect any change in H3K27ac that could explain an increase in *Kit* expression? Isn't an enhancer adoption mechanism that is not happening in this case? Which is completely independent from a change in H3K27ac-marked enhancer region.

Thank you for pointing this out. We acknowledge that the sentence is incorrect. We meant that H3K27ac enrichment is possible at an active gene promoter, but

we did not detect *Kit* activation in mutant MEFs. To avoid the confusion we have revised the text to provide a clearer description of H3K27ac in MEFs:

“But for an enhancer hijacking, H3K27ac enriched regions upstream to Pdgfra remained insulated (Fig.3f), and consequently there was no increase in Kit transcriptional activity in any of the mutant MEFs.”

6) In Supplementary Fig 4, the CTCF and H3K27ac ChIP-seq profiles are completely different between the Wt and the Kit Delta30k+ mutant. Can the authors explain this?

Thank you for noting. The difference between Wt and Kit Δ 30k+ ChIP-seq data in the Supplementary Fig.4 was a technical problem. We performed an independent sample preparation and further analysis revealed H3K27ac enrichment to be similar between the two groups. We included new data in the modified version of the figure (current Supplementary Fig.6).

7) The paragraphs “Generated mouse strains carrying boundary CBSs deletions” and “Eliminating the contribution of distance decrease to the Hi-C interaction profiles in mutant cells” can be removed and the manuscript structure modified by including the data within these two paragraphs in the other paragraphs. This structure produces redundancy in the text between the paragraphs. It also makes the reader wonder about some results, but those come only later on (just due to the current structure). Talking about the Delta2kb deletion with its full results than the Delta60kb with its full results in comparison to Delta2Kb...etc. would make the manuscript much easier to follow and give a nice structure to the manuscript.

Thank you for the suggestion. We have revised the manuscript to address your concerns.

We removed the paragraph “Generated mouse strains carrying boundary CBSs deletions” and restructured the text so that description of the deletions first appear in the paragraph of the correspondent TAD border: for *Pdgfra*/*Kit* border — “*Pdgfra* gene localisation at the TAD border challenges the border disruption”; for *Kit*/*Kdr* border — “*Kit*/*Kdr* boundary CBSs deletions result in the TADs fusion”.

We removed the paragraph “Eliminating the contribution of distance decrease to the Hi-C interaction profiles in mutant cells” by converting it to the Supplementary note 1 and partially including in the paragraph “*Kit*/*Kdr* boundary CBSs deletions result in the TADs fusion”.

“The establishment of inter-TAD interactions in mast cells and melanocytes was further confirmed by the subtraction maps of mutant and Wt cells with simulated deletions (Fig.4e,f). This approach was used to estimate the contribution of distance decrease on the rewiring of chromatin contacts and is discussed in more detail in the Supplementary Notes and demonstrated in Supplementary Fig.7.”

We also restructured the narration about Pdgfra Δ 2k and Pdgfra Δ 60k deletions. A description of a deletion and its effect on the 3D genome organization is grouped to Pdgfra Δ 2k first and Pdgfra Δ 60k second.

We believe that the changes improved the manuscript greatly. We hope you find the text structure to be more logical and clear.

8) The authors stipulate that the sub-structure of the Kit TAD into two self-interacting regions in the melanocytes and mast cells can be explained by the high-level transcription of the Kit gene. Can the authors block the transcription of the Kit gene by introducing a polyA terminating signal, and study the impact on 3D chromatin folding as well the effect on gene expression upon TAD border deletion.

Thank you for your suggestion. Conducting such an experiment would indeed be valuable to highlight the role of gene expression in the 3D organization of the Kit locus. However, genome editing in primary cells presents significant challenges due to low transfection efficiency. Consequently, we opted to use an immortalised cell line with a similar locus organization. Our search led us to murine mastocytoma cells P815, which exhibit expression levels of the Kit and Kdr genes comparable to our mast cells. Additionally, these cells show a profile of active chromatin marks (H3K27ac) closely resembling that observed in our samples.

After selecting suitable transfection conditions for these cells, we achieved a satisfactory efficiency rate of approximately 50%. Paradoxically, despite this efficiency, we were unable to produce deletions or even indels in these cells following CRISPR/Cas9 transfection. This outcome suggests a possible unique aspect in the functioning of the double-strand break repair system within these cells. Consequently, we currently lack a robust experimental model to conduct the proposed experiment.

9) Supplementary Fig 11 is a result that should be included in the results part and not shown and discussed in the discussion part.

Thank you for the suggestion. We included the result related to Kit Δ 300k deletion to the main Fig.5c (initially Supplementary Fig.11). We have revised the manuscript accordingly to discuss the result in more detail.

“Surprisingly, decrease of genomic distance between Kit and Kdr genes was also insufficient for Kdr activation in mast cells. We had performed transcriptome analysis on mast cells from a mouse strain carrying a 300 kb deletion at the Kit/Kdr TADs border, previously obtained in our laboratory⁴¹ (Kit Δ 300k, Fig.2a). We found an abnormal transcription in Kit Δ 300k that could be explained by transcription started from the Kit promoter, which remained intact after the deletion, and continued to the closest termination site within Kdr (Fig.5c). Of note, since Kdr locates on an opposing DNA strand to Kit, its transcript was antisense. The Kit Δ 300k mice strain is heterozygous, therefore Kit was expressed from a normal allele. To compare effects of Kit Δ 300k and Kit Δ 30k deletions on 3D genome organization would be of interest, but unfortunately the Kit Δ 300k is recessive lethal, so we were unable to obtain homozygous cells that are required to detect 3D genome changes.

In Kit Δ 300k we did not detect a Kdr sense transcript, which potentially could be explained by the fact that an antisense transcript could inactivate a sense transcription through interference (see review⁴²). But nor did we detect increased activity of genes downstream to Kdr which were unaffected by the antisense transcription. Thus, we believe that a high level of Kit transcription in mast cells could contribute to restriction of an enhancer hijacking rather than genomic distance.”

Minor comments:

1) It is not clear in Fig 2 which CTCF sites are disrupted by the 2kb deletion, a magnification showing clearly this is necessary or adding a supplementary figure showing clearly this is necessary.

Thank you for pointing this out. We updated Fig.2 with a magnification of the Pdgfra/Kit border for a better view on the CTCF sites deleted in the Pdgfra Δ 2k, as well as in other introduced deletions. Please see the updated version. We hope it clarifies the confusion.

2) Line 147: correct please. With two sites in forward and one in reverse orientation.

Thank you very much for your comment, we have corrected it: *“The Kit/Kdr boundary is a joint between two adjacent TADs which contains three clustered CBSs, with two sites in forward and one in reverse orientation”.*

3) Lines 176-177: correct please. (Supplementary Figure 7)

Thank you for your comment. We have modified figures to include subtraction maps in the main figures rather than supplementary to provide an easier way to follow the results. We have revised the manuscript according to the changes in reference of figures.

4) Lines 210-211: correct please. Kit with its upstream enhancers are located within the left half of the TAD and thus are insulated from the right half.

Thank you very much for your comment, we have corrected it: *“Kit with its upstream enhancers are located within the centromeric half of the TAD and thus are insulated from the telomeric half”*.

5) It would be nice to restructure panels c, d, and e from Fig 4 to have the Wt, Kit Delta30k, and the subtraction Hi-C maps in mast cells in one panel, and in parallel the same Hi-C maps in melanocytes. This will make the Figure easy to follow. The same should apply for Fig 3. It would be nice to include the subtraction maps in the main figure to follow easily the results section.

Thank you for your suggestion. We modified Fig.3 and Fig.4 so that Hi-C maps of one cell type would be in one panel. We also included subtraction maps into the main figures from supplementary. We believe it improved the readability of the figures greatly.

6) Line 215: please correct. However, we noted that contacts rewiring mainly involved the left half of the Kit TAD.

Thank you very much for your comment, but changes in contacts were observed in the telomeric half (right) of the Kit TAD, but not in centromeric (left) half due to the Kit insulation. We have corrected it: *“However, we noted that contacts rewiring mainly involved the telomeric half of the Kit TAD, which merged with the Kdr TAD”*.

7) In Fig 4d, there is no H3K27ac for the Wt. However, this is present for the mutant. There is no CTCF data for both the Wt and mutant. Can the authors explain this?

Thank you for the notion. We did include H3K27ac ChIP-seq for both Wt and mutant melanocytes in the modified Fig.4d.

We agree that lack of CTCF data for melanocytes is a limitation of our study. The issue was also raised by Reviewer 3 in major comment 1. Here we cite our answer:

“Unfortunately we faced technical difficulties with preparation of the samples for ChIP-seq. Taking into account the fact that production of a primary melanocytes culture requires 8 decapitated 4-day old mouse pups and yields in only 200,000 cells after a month of cultivation we argue that it is unethical to proceed with the optimisation of CTCF ChIP-seq that demands at least two replicas for an analysis.

We believe that there are two main issues caused by our decision. How can we know that there is a CTCF binding site in the Kit gene in melanocytes? How can we know that the introduced deletions did occur and the expected CTCF binding motifs were in fact deleted in melanocytes?

For the Kit gene, though CTCF binding may vary between cell types, analyzing data from several cell types together with Hi-C map of melanocytes could provide an understanding of CTCF binding in melanocytes. To tackle the issue we addressed CTCF ChIP-seq for human melanoma cell lines (uveal melanoma patient-derived xenografts, SK-MEL-147, COLO829, M14), since there was no available data for mouse melanocytes or melanoma cells, and available mouse cell types (MEFs, mast cells, mESCs). All the described data demonstrates a CTCF binding signal at the Kit first exon (Fig.3 here). Considering this together with our Hi-C maps from mouse melanocytes where a distinct TAD border is visible rising from the Kit gene we believe it is a CTCF-mediated spatial feature.

For the deletions, we introduced them on an organismal level which allows us to make a conclusion about deletion in melanocytes based on other evidence for the same genotype. Firstly, to establish mouse strains we implemented PCR genotyping using DNA from mouse tails, so we confirmed animals to carry the Kit Δ 30k deletion. Secondly, CTCF ChIP-seq data of mast cells from Kit Δ 30k mice revealed a lack of CTCF binding motifs corresponding to the introduced deletion Kit Δ 30k. Thus, we can conclude with a certainty that the introduced deletion was present in melanocytes.

We acknowledge that CTCF ChIP-seq is a crucial part of melanocytes characterisation in our study. But we hope that the additional information provided here can account for it to some degree.”

8) Line 232: please correct and replace “left” by “right”.

Thank you for your comment. We have corrected and to minimize the confusion we introduced the terms centromeric/telomeric rather than left/right:

“The inner structure of the Kit TAD was divided with Kit and its downstream enhancers being insulated from the centromeric part of the TAD (Fig.4d).”

9) The results presented in Supplementary Fig 7b for the Delta60k mutant contradict the title “Pdgfra/Kit CBSs deletions were insufficient to change the regulatory and architectural patterns”, as there is a gain of chromatin contacts in this mutant. However, this is insufficient to drive ectopic gene activation.

Thank you for noting this. We agree that the title of the results from CBSs deletions at the Pdgfra/Kit TADs border is misleading. Deletions at the Pdgfra/Kit TAD border did result in gain of chromatin contacts, though the effect was marginal and TADs remained largely separate. We reconsidered the title to emphasize the inability of the deletions we introduced to result in Pdgfra and Kit TADs fusion accompanied by ectopic gene activation: *“Pdgfra gene localisation at the TAD border challenges the border disruption”*.

10)Supplementary Fig 10 is not cited in the text.

Thank you for your comment. We have to disagree because Supplementary Fig.10 was cited in the text, lines 286-289:

“We performed a manual counting of melanocytes on immunostained skin sections from 4-day-old pups aiming to explain the specific phenotype of the paler Kit Δ 30k mice (Supplementary Fig.10).”

However in the modified manuscript we have replaced the figure by more representative images of the skin sections which can be found in the Supplementary Fig.8. We have revised figure citations in the text accordingly.

11)Fig 5c is unnecessary as there is no corresponding Wt with the same magnification.

Thank you very much for your comment, we removed the picture.

12) The authors should be careful when citing the Figures in the text. It is not possible to cite Fig 4c, d, e before citing Fig 4a, b. This applies to the full manuscript.

Thank you for noting it. We have revised the manuscript to ensure the citations of the figures are introduced sequentially according to their placement in figures.

Reviewer #3 (Remarks to the Author):

In this manuscript, the authors describe the regulatory landscape and TAD structure at the extended locus of *Pdgfra*, *Kit*, and *Kdr* genes. Comparison of the locus TAD structure in several species revealed the preservation of TAD integrity in tetrapods. They then created multiple genomic deletions in mice that involve CTCF-binding sites at TAD boundaries that separate the regulatory information for these genes. The consequences of CTCF-binding site deletions were analysed in different cellular contexts of cultured cells, reflecting several on/off states (or intermediate states) in gene expression and chromatin state at the locus. Effects on genome folding were assessed by capture-HiC and changes in the transcriptome by RNA-seq. In general, this is an interesting study, which adds to a growing list of *in vivo* studies investigating the consequences of TAD boundary deletions on gene expression. The authors highlight that the choice of the cellular context is important to appreciate the regulatory effects upon CTCF-binding site deletions associated with TAD boundaries. However, I have several concerns which diminish my enthusiasm. Some conclusions appear not to be supported by the data or data are not sufficiently described and presented. Furthermore, the conclusions and their discussion are rather limited. Below I explain my concerns and reservations I have about the manuscript.

Some conclusions seem not to be supported by the data.

1) First, CTCF ChIP-seq in Melanocytes wild-type and mutant cells are not presented. Lines 343 to 344 state “Notably, the border of an insulated region coincides with a CTCF site at the *Kit* promoter in melanocytes, whereas it shifts toward the *Kit* terminator in mast cells.”. However, no CTCF ChIP-seq data are shown or provided by the authors to further assess and compare the described differential CTCF binding at the *Kit* promoter and gene body. This data set is also important to evaluate some of the author’s conclusions that internal *Kit* TAD insulation is provided by an epigenetic environment (see lines 241 to 243).

Thank you for your comment. We agree that lack of CTCF ChIP-seq in melanocytes is a disadvantage of our study. Unfortunately we faced technical difficulties with preparation of the samples for ChIP-seq. Taking into account the

fact that production of a primary melanocytes culture requires 8 decapitated 4-day old mouse pups and yields in only 200,000 cells after a month of cultivation we argue that it is unethical to proceed with the optimisation of CTCF ChIP-seq that demands at least two replicas for an analysis.

We believe that there are two main issues caused by our decision. How can we know that there is a CTCF binding site in the *Kit* gene in melanocytes? How can we know that the introduced deletions did occur and the expected CTCF binding motifs were in fact deleted in melanocytes?

For the *Kit* gene, though CTCF binding may vary between cell types, analyzing data from several cell types together with Hi-C map of melanocytes could provide an understanding of CTCF binding in melanocytes. To tackle the issue we addressed CTCF ChIP-seq for human melanoma cell lines (uveal melanoma patient-derived xenografts, SK-MEL-147, COLO829, M14), since there was no available data for mouse melanocytes or melanoma cells, and available mouse cell types (MEFs, mast cells, mESCs). All the described data demonstrates a CTCF binding signal at the *Kit* first exon (Fig.3 here). Considering this together with our Hi-C maps from mouse melanocytes where a distinct TAD border is visible rising from the *Kit* gene we conclude it is a CTCF-mediated spatial feature. We believe that the possibility to gain information about CTCF binding at the *Kit* gene from other sources resolves the ethical problem.

For the deletions, we introduced them on an organismal level which allows us to make a conclusion about deletion in melanocytes based on other evidence for the same genotype. Firstly, to establish mouse strains we implemented PCR genotyping using DNA from mouse tails, so we confirmed animals to carry the *Kit* Δ 30k deletion. Secondly, CTCF ChIP-seq data of mast cells from *Kit* Δ 30k mice revealed a lack of CTCF binding motifs corresponding to the introduced deletion *Kit* Δ 30k. Thus, we can conclude with a certainty that the introduced deletion was present in melanocytes.

We acknowledge that CTCF ChIP-seq is a crucial part of melanocytes characterisation in our study. But we hope that the additional information provided here can account for it to some degree.

Fig. 3. CTCF binding at the *Kit* promoter. CTCF ChIP-seq data for human melanoma (**a**) and mouse cells (**b**). Uveal melanoma patient-derived xenografts (PDX) is from Gentien D. et al., 2023¹⁸; SK-MEL-147 is from Carcamo S. et al., 2022¹⁹; COLO829 is from Poulos R. C. et al., 2016²⁰; M14 is from Chu Z. et al., 2022²¹; mESCs is from Justice M., 2020²². MEFs and mast cells data is from the current study.

2) Second, the results from the hybrid experiments, conducted to distinguish between trans and cis effects, do not align with the manuscript text description and are contradicting. Lines 313 to 314, “We have shown that almost all (98%) Kdr transcripts in hybrid melanocytes originate from the chromosome carrying the *Kit* Δ30k deletion (Fig.5g).” While samples from wild-type hybrid cells show equal expression of Kdr from *M. mus* and *M. cas* alleles in Figure 5g, the hybrids cells with the mutant allele show exclusive expression from *M. cas* allele but not *M. mus* allele that carries the

Kit Δ 30k deletion. This contradicts the conclusion in the text. Why is Kdr expression higher from M.cas allele in hybrids with mutant allele compared to Kdr expression from M.cas allele in wild-type hybrids? Why do the authors detect Kdr expression from M.mus allele in wild-type hybrids but not from M.mus allele carrying the Kit Δ 30k deletion? Furthermore, no respective statistical test is shown in Figure 5 g as described in the text, line 315, “The overall level of Kdr expression was significantly lower in the control experiment with M. castaneus x M. musculus Wt hybrids, with both alleles being equally active.”

Thank you for noting that the alleles caption on the figure got swapped. The information provided in the text description (98% of the Kdr transcripts in hybrid melanocytes originate from the chromosome carrying the Kit Δ 30k deletion) is what we detect from our RNA-seq data analysis and what we intended to illustrate by the figure. However, inspired by your suggestion, we have changed the analysis method to better emphasize the difference between wild-type and Kit Δ 30k allele. Instead of counting the proportion of reads carrying SNPs from one of the two species, we have compared normalized read counts from wild-type and Kit Δ 30k, using read counts from the M.cas allele as a normalization factor. The significance of differences was confirmed by the Mann-Whitney test. Please see Fig.5e.

“We isolated primary cells from the epidermis of the hybrids, differentiated them, performed RNA-seq in melanocytes, and calculated the read counts from one of the two species based on the SNP analyses. Then, we used read counts from the M. castaneus allele as a normalization factor and compared the level of Pdgfra, Kit and Kdr expression between wild-type and Kit Δ 30k alleles.”

3) Last, the authors describe differential subTAD structure for Kit TAD in mast cells and melanocytes. Different localization of Kit enhancers might account for intra-TAD insulation (as described in lines 112-114, “According to our H3K27Ac ChIP-seq data (schemes at Fig.1 and Fig.3-4), Pdgfra and Kit enhancers localize upstream of the genes in MEFs and mast cells, respectively. In melanocytes, by contrast, the H3K27ac-rich enhancer region is located downstream.”). A major conclusion derived from the manuscript is that the relative position of active enhancers and gene transcription within the subTAD structure determines the insulation defects and gene misexpression in TAD fusion events with Kdr TAD. H3K27ac data for Kit delta30k is shown in Figure 4 but not in wild-type melanocytes. The deposited manuscript data entry shows “melano_h3k27ac_wt.fc.signal.bw” but the data is not provided in the manuscript. In addition, for better appreciation of the schematics in Figures 1 b,c, and d, depicting the “H3K27ac-rich enhancer region”, it is highly advisable to cross-reference

the depicted schematic enhancer regions in Figure 1 with the actual data figures (e.g. by highlighting enhancer regions in Figure 4).

Thank you for noticing. We did not intend to cause confusion by a missing H3K27ac in the figure. We included H3K27ac for wild-type melanocytes in Fig.4. To provide an easier readability of the manuscript we highlighted regions that we refer to as enhancers in Fig.1c. We hope the modifications provide clarity to the manuscript.

Addition comments:

1) In both, mast cells and melanocytes, Kit is highly expressed, but the removal of the TAD border leads to ectopic activation of the Kdr gene in melanocytes only. The authors reasonably discuss and correlate Kdr misexpression with the distribution and accessibility to potential enhancers within the Kit subTADs. In wild-type mast cells Kdr seems not to be expressed and therefore the Kdr promoter might be not permissive to regulatory changes (enhancer adoption) induced by the Kit Δ 30k+ deletion. What is known about Kdr repression (repressive chromatin e.g., H3K27me3) in the investigated cell types or related cell types? Could the repressive chromatin state at the Kdr promoter account for the absence of gene misexpression. Similarly, a repressive chromatin state could account for the absence of gene misexpression of Kit in the Pdgfra Δ 60k mutants. A similar discussion as described above for Kdr could enrich the discussion and conclusions.

Thank you for raising an interesting topic. We do think that *Kdr* accessibility to an enhancer adoption could contribute to its activation in melanocytes and not mast cells. A repressive state of *Kdr* is confirmed by H3K27me3 data from Berrozpe G. et al., 2017 (Supplementary Fig.2 in the modified manuscript). Moreover, all the region from the *Kit* termination site to the closest active gene *Srd5a3* appears to be enriched in H3K27me3. Our RNA-seq data for wild-type revealed that *Kdr* is expressed in melanocytes at a low level, but it is silent in mast cells. Thus, *Kdr* is repressed in mast cells, unlike melanocytes.

However, according to our Hi-C maps, spatial interactions between *Kit* and *Kdr* were not established in mast cells. Since the enhancers are located upstream to *Kit* in mast cells, it suggests that *Kit* enhancers remained insulated from *Kdr*. Whether the repressive state of *Kdr* could influence the spatial insulation is unclear, as well as the causation between gene activity and epigenetic state. The main possible explanation for a tissue-specific *Kdr* activation we consider to be the difference in *Kit* transcription level (in mast cells *Kit* is tenfold more

transcribed compared to melanocytes), for RNAPII was revealed to act as a barrier for loop extrusion⁴.

As for the absence of misexpression of *Kit* in the *Pdgfra* $\Delta 60k$, *Kit* appears to be enriched in H3K27me3 in MEFs according to data from Phan Q. M. et al., 2023 (Supplementary Fig.2 in the modified manuscript). Though we consider the disruption of *Pdgfra/Kit* TADs incomplete, because we revealed an increase of CTCF binding in the more internal CTCF motif that could account for the rescue of the *Pdgfra* TAD border (Supplementary Fig.4 in the modified manuscript). Thus, enhancer hijacking was restricted by insulation between *Pdgfra* and *Kit* TADs.

According to your suggestion, we have revised the manuscript to discuss the problem in more detail:

*“Of note, for mast cells data from another study⁴⁰ reveals that all the region from the *Kit* termination site to the closest active gene *Srd5a3* appears to be enriched in H3K27me3, suggesting a repressive state of *Kdr* (Supplementary Fig.2a).”*

*“We found *Kdr* to be expressed at a low level in *Wt* melanocytes, and silent in *Wt* mast cells. Taking into account *Kdr* enrichment of a repressive epigenetic mark H3K27me3 in mast cells, it is possible that the repressive state of *Kdr* makes it inaccessible to an enhancer hijacking. Our Hi-C maps revealed that spatial interactions between *Kit* and *Kdr* were not established in mast cells, and the active enhancers remained insulated from *Kdr*. However, whether the repressive state of *Kdr* could influence the spatial insulation is unclear, as well as the causation between gene activity and epigenetic state.”*

2) The H3K27ac ChIP-seq track in *Kit* $\Delta 30k+$ mast cells in Supplementary Figure 4 b shows almost a background signal. Could the authors explain the reduced signal compared to the wild type? Is this a technical problem of the ChIP or a consequence of the mutation? This needs to be clarified to fully comprehend the results.

Thank you for pointing this out. The reduced H3K27ac signal of the *Kit* $\Delta 30k+$ mast cells relative to wild type was in fact due to a technical problem of the ChIP. To clarify the issue we performed an independent preparation of a H3K27ac ChIP-seq on *Kit* $\Delta 30k+$ mast cells and the resulting data was not as different from the wild type data. We updated the Supplementary Figure 4 with new data.

- 3) Figure 1a depicts relative gene expression by gene ranking (in %). This representation does not allow a comparison of gene expression between cell types or an understanding of quantitative assessments of gene expression as done by the authors in the text. E.g. lines 202 to 203 “Kdr is normally inactive in mast cells and is marginally expressed in melanocytes (Fig.1a).” or lines 344 to 345 “Though Kit is expressed in both cell types, the expression level in mast cells is 10 times higher.” At least a graph or table should be included in the supplementary data, comparing normalized counts (e.g. transcripts per million) of relevant genes at the locus and between tissues. In addition, gene expression levels for all the mutant alleles should be incorporated into such a data table or figure. How do transcriptional changes in Kit Δ 30K and Kit Δ 30K+ compare to each other? Does the increased ectopic signal in the Kit Δ 30K+ deletion allele results in more pronounced Kdr misexpression?

Thank you for your suggestion. We have taken your advice and modified the figure to show TPM (Transcripts Per Million) expression of the Pdgfra, Kit, and Kdr genes in wild-type fibroblasts, mast cells, and melanocytes. These violin plots actually better reflect gene activity in different cell types, and in addition, they allow comparison of activity levels between cell types.

We did not display the level of gene activity for mutant lines in this figure in order not to disrupt the flow of the story. However, in the revision, we have changed the text when describing the analysis of gene activity. This is to emphasize that, in addition to the Kdr gene in melanocytes, other genes within \pm 1Mb range around the deletion do not change significantly.

We did not obtain melanocytes from the Kit Δ 30K+ line, so we cannot answer whether the additional deletion of the CTCF recognition sites contributes to the increase in Kdr gene activity.

- 4) The display of CTCF binding sites in Figure 2 does not match the text description. Lines 132 to 134, “Pdgfra TAD border is characterized by three strong CBSs in reverse orientation, located within the Pdgfra gene body (Fig.2c). The Pdgfra Δ 2k strain carries a 2 kb deletion of the three CTCF peaks at the Pdgfra TAD border.” First, Figure 2 shows three CTCF motifs in reverse orientation in the Pdgfra gene body. However, Supplementary Figure 3 and the magnification to a 7 kb region (supposedly comprising the Pdgfra promoter) shows CTCF binding and motif orientation of two reverse and one forward-facing CTCF motif and a further distally located CTCF binding site with reverse orientation. Second, a 2 kb deletion does not match the described and displayed genomic size of three CTCF binding sites in reverse orientation. Which CTCF binding sites are deleted in Pdgfra

Δ 2k allele? Furthermore, lines 140 to 142, "...*Pdgfra* Δ 60k strain carries a 60 kb deletion spanning the whole *Pdgfra*/Kit TAD boundary region, and therefore removing the *Pdgfra* coding sequence, leaving only 3 out of 22 exons and a promoter." Can the authors clarify the overlap of *Pdgfra* Δ 60k and *Pdgfra* Δ 2k deletion? Does the *Pdgfra* Δ 60k allele delete promoter proximal CTCF binding sites (Magnification in Suppl. Figure 3) although the deletion allele leaves the promoter region and three exons intact as described in the text?

Thank you for noting that text and figures of the CTCF binding sites are misleading. We did not intend to cause confusion.

We have revised the text to clarify the misleading information.

*"The *Pdgfra* border consists of three strong CBSs, one forward and two reverse-facing, that overlap the *Pdgfra* gene body (Fig.2a,b). The *Kit* border consists of one forward-facing CBS."*

We have modified Fig.2 to better visualize how CTCF motifs, *Pdgfra* gene, and corresponding deletions are located relative to each other. Since it resembles too much of Supplementary Fig.3, we have deleted Supplementary Fig.3 to avoid confusion. So Fig.2 now is a united version of previous Fig.2 and Supplementary Fig.3.

Previously the sizes of the *Pdgfra* Δ 2k deletion and the *Pdgfra* TAD border seemed to not match, because we provided magnification of a larger region, than the deletion (7 kb, including 2 kb of the border and 5 kb of CTCF motifs located further inside the TAD) in the Supplementary Fig.3. To fix the problem we replaced it with a magnification of the whole border region in the updated version of Fig.2. We hope it is clear now that the three CTCF binding sites are deleted in the *Pdgfra* Δ 2k and the deletion is 2 kb.

According to your suggestion, we updated Fig.2 to better visualize how *Pdgfra* Δ 2k and *Pdgfra* Δ 60k correspond to each other. Both deletions start the same, but *Pdgfra* Δ 60k further continues to the *Kit* side of the border. Consequently, *Pdgfra* Δ 2k is at the intron region and does not disrupt the coding region of *Pdgfra*, while *Pdgfra* Δ 60k affects the coding region and only the promoter and the first exon of the *Pdgfra* gene remains intact.

We believe that the modifications improved our manuscript and we hope the confusion is clarified.

- 5) The authors should revise the manuscript for the description of CTCF motif orientation. For instance, in lines 146 to 147, “The Kit/Kdr boundary is a joint between two adjacent TADs which contains three clustered CBSs, with two sites in reverse and one in forward orientation (Fig.2c).” According to the definition in the previous manuscript text, Supplementary Figure 3 description, and display of CTCF motif orientation the Kit/Kdr boundary would consist of one reverse and two forward-oriented CTCF motifs.

Thank you very much for your comment, we have corrected it:

“The Kit/Kdr boundary is a joint between two adjacent TADs which contains three clustered CBSs, with two sites in forward and one in reverse orientation (Fig.2a,c).”

- 6) Lines 209 to 211, “Contact maps from the wild-type mast cells denoted a nested substructure of the Kit TAD, with two self-interacting regions. Kit with its upstream enhancers are located within the right half of the TAD and thus are insulated from the left half (Fig.1c).” and 216 to 217 “The insulation inside the Kit TAD was not affected by the deletion, thus, a self-interacting region, located at the right side, remained isolated.” The description of “right half” or “right side” does not match schematic in Figure 1c for mast cells. According to Figure 1 c, enhancers in mast cells are located in the left half of Kit TAD. Is this the case? See also line 232 and 235, “left half” instead of right half? In general, the description of genomic orientation and position is not consistent in the manuscript, using terms such as right, left, upstream, or downstream of different genes. The readability of the manuscript should be improved by using consistent formatting for relative positions. For instance, referring to relative positions by using the centromeric and telomeric orientation of the chromosome (or similar).

Thank you for your note, we have corrected the text by using centromeric and telomeric orientation:

“Contact maps from the Wt mast cells denoted a nested substructure of the Kit TAD, with two self-interacting regions. Kit with its upstream enhancers are located within the centromeric half of the TAD and thus are insulated from the telomeric half (Fig.4c).”

“The insulation inside the Kit TAD was not affected by the deletion, and a self-interacting region, located at the centromeric side, remained isolated.”

- 7) Line 190 to 192, “In *Pdgfra* Δ 60k, an intergenic region between *Pdgfra* and *Kit* is transcribed, starting from the intact *Pdgfra* promoter and continuing

to the nearest termination site.” Could the authors highlight and/or enlarge the region in the respective figures? It is not clear to the reader what the authors are referring to.

Thank you for the notion. We provided an additional supplementary figure, where the new transcript is enlarged, to clarify what we are referring to. Please see Supplementary Fig.3.

8) In the discussion the authors describe a larger 300kb deletion, involving the Kit/Kdr border, which has no gene expression consequences on Kdr. However, the description lacks a deeper relation to the produced data and conclusions. According to the model presented in Supplementary Figure 12 (Mast cells), the 300 kb deletion removes the internal Kit TAD border and therefore brings Kdr in close proximity to potential mast cell active enhancers. Could the authors describe why they think that the direct proximity of Kdr to these enhancers has no consequences? For instance, to what extent is the Kdr promoter permissive to the regulatory input (of Kit landscape) in mast cells?

Thank you for bringing to our attention that we overlooked an interesting result about Kit Δ 300k deletion, which decreases genomic distance between *Kit* and *Kdr*.

In the previous question we discussed two factors that could contribute to inability for an enhancer adoption by *Kdr* in mast cells, that is (1) *Kdr* is repressed in mast cells and (2) *Kit* high transcription level could contribute to robustness of insulation. Here we cite our answer to the previous question:

“We do think that Kdr accessibility to an enhancer adoption could contribute to its activation in melanocytes and not mast cells. A repressive state of Kdr is confirmed by H3K27me3 data from Berrozpe G. et al., 2017 (Supplementary Fig.2 in the modified manuscript). Moreover, all the region from the Kit termination site to the closest active gene Srd5a3 appears to be enriched in H3K27me3. Our RNA-seq data for wild-type revealed that Kdr is expressed in melanocytes at a low level, but it is silent in mast cells. Thus, Kdr is repressed in mast cells, unlike melanocytes.

However, according to our Hi-C maps, spatial interactions between Kit and Kdr were not established in mast cells. Since the enhancers are located upstream to Kit in mast cells, it suggests that Kit enhancers remained insulated from Kdr. Whether the repressive state of Kdr could influence the spatial insulation is unclear, as well as the causation between gene activity and epigenetic state. The main possible explanation for a tissue-specific

Kdr activation we consider to be the difference in Kit transcription level (in mast cells Kit is tenfold more transcribed compared to melanocytes), for RNAPII was revealed to act as a barrier for loop extrusion⁴.”

We believe that the same reasoning might be true for Kit Δ 300k, since we analyzed these deletions in the same cell type, mast cells.

An additional feature of the Kit Δ 300k is the appearance of an antisense *Kdr* transcript (Fig.5 of the modified manuscript). Kit Δ 300k deleted most of the *Kit* coding region except promoter and the first exon. So transcription was able to start from *Kit* and continue to the closest termination site. Of note, RNA-seq revealed *Kit* transcript because Kit Δ 300k is a heterozygous mouse strain, and *Kit* was able to be transcribed from a normal allele. As *Kit* is a forward orientated gene and *Kdr* is a reverse orientated gene, the transcription of *Kdr* coding region was antisense. It is possible for an antisense transcript to block a sense transcription, which potentially could account for the lack of *Kdr* sense transcript despite decrease of genomic distance to active enhancers. However, we did not detect an increase in activity of more proximal genes, located downstream to *Kdr*, which were unaffected by antisense transcripts.

We updated the manuscript to better present the result. We have included a scheme of Kit Δ 300k deletion relative to other deletions in the locus (modified Fig.2), and incorporated RNA-seq analysis of Kit Δ 300k mast cells to the main figure from supplementary (modified Fig.5c). We have revised the text to describe the result in more detail in Results section:

“Surprisingly, decrease of genomic distance between Kit and Kdr genes was also insufficient for Kdr activation in mast cells. We had performed transcriptome analysis on mast cells from a mouse strain carrying a 300 kb deletion at the Kit/Kdr TADs border, previously obtained in our laboratory⁴¹ (Kit Δ 300k, Fig.2a). We found an abnormal transcription in Kit Δ 300k that could be explained by transcription started from the Kit promoter, which remained intact after the deletion, and continued to the closest termination site within Kdr (Fig.5c). Of note, since Kdr locates on an opposing DNA strand to Kit, its transcript was antisense. The Kit Δ 300k mice strain is heterozygous, therefore Kit was expressed from a normal allele. To compare effects of Kit Δ 300k and Kit Δ 30k deletions on 3D genome organization would be of interest, but unfortunately the Kit Δ 300k is recessive lethal, so we were unable to obtain homozygous cells that are required to detect 3D genome changes.

In Kit Δ 300k we did not detect a Kdr sense transcript, which potentially could be explained by the fact that an antisense transcript could inactivate a sense transcription through interference (see review⁴²). But nor did we detect increased activity of genes downstream to Kdr which

were unaffected by the antisense transcription. Thus, we believe that a high level of Kit transcription in mast cells could contribute to restriction of an enhancer hijacking rather than genomic distance.”

We conclude that in case of Kit/Kdr TADs border disruption in mast cells, genomic distance might not be the main factor to regulate insulation to an enhancer adoption.

9) Line 289 says “However, the analysis did not reveal significant differences between two genotypes;” and the description of Supplementary Figure 10 refers to the quantification of Kit-positive cells in melanocytes. No quantification is shown in the manuscript. In the current state Supplementary Figure 10 does not provide information to the reader.

Thank you for this remark. In the revised version of our manuscript we have decided to eliminate this part of the narrative as uninformative.

10)The terms like “conservativity” and “conservatism” or “since TADs are conservative” (line 435) are not appropriate word choices to describe conserved relative TAD position and integrity between species.

Thank you for bringing this to our attention. We do agree that the term “conservativity” relative to TADs demands careful usage. To emphasize this we revised the text to mention the contradiction and we introduced a limitation to what we refer to as conservative:

*“Consequently, we introduce a limitation and by conservativity we understand that *Pdgfra*, *Kit*, and *Kdr* are (1) in close genomic proximity, (2) each located within a separate TAD”.*

11)In lines 176 to 177. Wrong figure reference. Supplementary Fig.4 and Fig.5a,b both refer to Kit Δ 30+ allele but not *Pdgfra* Δ alleles. A likely reference is Supplementary Figure 7?

Thank you very much for your comment. It was in fact a reference to Supplementary Fig.7. However for easier readability we modified the manuscript to implement subtraction maps from Supplementary Fig.7 into a main figure of a corresponding deletion, and we removed the previous Supplementary Fig.7. We revised figure citations accordingly.

12)Wrong Figure reference in line 220. Are the authors referring instead to Supplementary Figure 7?

Thank you for noting. We were referring to Supplementary Fig.7. However for easier readability we included a subtraction map for Kit Δ 30k+ from Supplementary Fig.7 to the figure of Kit Δ 30k+ characterisation. Please see Supplementary Fig.6 for a modified version. We revised figure citations accordingly.

13)Wrong Figure reference in line 280. Are the authors referring instead to Supplementary Figure 8?

Thank you for your comment. In the revised version of our manuscript the Supplementary figures and the text were reorganized and the aforementioned figures are no longer present.

References

1. Botten, G. *et al.* Structural Variation Cooperates with Permissive Chromatin to Control Enhancer Hijacking-Mediated Oncogenic Transcription. *Blood* **142**, 336–351 (2023).
2. Ringel, A. R. *et al.* Repression and 3D-restructuring resolves regulatory conflicts in evolutionarily rearranged genomes. *Cell* **185**, 3689-3704.e21 (2022).
3. Dixon, J. R. *et al.* Topological domains in mammalian genomes identified by analysis of chromatin interactions. *Nature* **485**, 376–380 (2012).
4. Banigan, E. J. *et al.* Transcription shapes 3D chromatin organization by interacting with loop extrusion. *Proceedings of the National Academy of Sciences* **120**, e2210480120 (2023).
5. Vierbuchen, T. *et al.* AP-1 Transcription Factors and the BAF Complex Mediate Signal-Dependent Enhancer Selection. *Molecular Cell* **68**, 1067-1082.e12 (2017).
6. Berrozpe, G. *et al.* A Distant Upstream Locus Control Region Is Critical for Expression of the Kit Receptor Gene in Mast Cells. *Mol Cell Biol* **26**, 5850–5860 (2006).

7. Li, Y. *et al.* GATA2 regulates mast cell identity and responsiveness to antigenic stimulation by promoting chromatin remodeling at super-enhancers. *Nat Commun* **12**, 494 (2021).
8. Calero-Nieto, F. J. *et al.* Key regulators control distinct transcriptional programmes in blood progenitor and mast cells. *The EMBO Journal* **33**, 1212–1226 (2014).
9. Berrozpe, G. *et al.* Polycomb Responds to Low Levels of Transcription. *Cell Reports* **20**, 785–793 (2017).
10. Infarinato, N. R. *et al.* BMP signaling: at the gate between activated melanocyte stem cells and differentiation. *Genes Dev.* **34**, 1713–1734 (2020).
11. Zhang, S. *et al.* H3K27ac nucleosomes facilitate HMGN localization at regulatory sites to modulate chromatin binding of transcription factors. *Commun Biol* **5**, 1–14 (2022).
12. Phan, Q. M. *et al.* Lineage commitment of dermal fibroblast progenitors is controlled by Kdm6b-mediated chromatin demethylation. *The EMBO Journal* **42**, e113880 (2023).
13. Lu, L. *et al.* Robust Hi-C Maps of Enhancer-Promoter Interactions Reveal the Function of Non-coding Genome in Neural Development and Diseases. *Molecular Cell* **79**, 521-534.e15 (2020).
14. Jacob, T. *et al.* Molecular and spatial landmarks of early mouse skin development. *Developmental Cell* **58**, 2140-2162.e5 (2023).
15. Paliou, C. *et al.* Preformed chromatin topology assists transcriptional robustness of Shh during limb development. *Proceedings of the National Academy of Sciences* **116**, 12390–12399 (2019).

16. Islam, Z. *et al.* Active enhancers strengthen insulation by RNA-mediated CTCF binding at chromatin domain boundaries. *Genome Res.* **33**, 1–17 (2023).
17. Oh, S. *et al.* Enhancer release and retargeting activates disease-susceptibility genes. *Nature* **595**, 735–740 (2021).
18. Gentien, D. *et al.* Multi-omics comparison of malignant and normal uveal melanocytes reveals molecular features of uveal melanoma. *Cell Reports* **42**, 113132 (2023).
19. Carcamo, S. *et al.* Altered BAF occupancy and transcription factor dynamics in PBAF-deficient melanoma. *Cell Rep* **39**, 110637 (2022).
20. Poulos, R. C. *et al.* Functional Mutations Form at CTCF-Cohesin Binding Sites in Melanoma Due to Uneven Nucleotide Excision Repair across the Motif. *Cell Reports* **17**, 2865–2872 (2016).
21. Chu, Z. *et al.* STAG2 regulates interferon signaling in melanoma via enhancer loop reprogramming. *Nat Commun* **13**, 1859 (2022).
22. Justice, M., Carico, Z. M., Stefan, H. C. & Downen, J. M. A WIZ/Cohesin/CTCF Complex Anchors DNA Loops to Define Gene Expression and Cell Identity. *Cell Reports* **31**, 107503 (2020).

REVIEWERS' COMMENTS

Reviewer #1 (Remarks to the Author):

Most of my comments have been addressed, which greatly improved the manuscript. There is one minor comment to the revised version. In the updated Fig. 5e, the y-axis is different between different genes. Because all of them are presumably scaled to the Cast allele, their y-axis should be 0-1 or 0-100%. Please revise them to make them comparable.

Reviewer #2 (Remarks to the Author):

The authors addressed my comments.

Reviewer #3 (Remarks to the Author):

I want to thank the authors for their thoughtful responses to the questions and concerns raised. The clarifications and introduced changes enhance the readability and potential significance of the manuscript.

Some suggestions for text editing:

Line 147-150.: Long sentence and the dependent clause, “, for a more proximal regulatory region was essential in melanocytes.”, is not well connected to the main sentence. I suggest splitting the sentence.

Line 252-253: Replace “We did not reveal...” with e.g. “The analysis did not reveal...”

Line 155 and line 375: Replace “... genomic data ...” with “... functional genomic data...” to put more emphasis on the type of data generated to capture the cell-type dynamics, i.e. ChIP-seq, RNA-seq, ChIP-seq.

REVIEWERS' COMMENTS

Reviewer #1 (Remarks to the Author):

Most of my comments have been addressed, which greatly improved the manuscript. There is one minor comment to the revised version. In the updated Fig. 5e, the y-axis is different between different genes. Because all of them are presumably scaled to the *Cast* allele, their y-axis should be 0-1 or 0-100%. Please revise them to make them comparable.

Thank you for the comment. We did not intend to cause confusion. The y-axis in the Fig.5e are not required to be in one range, because data scaled to the *Cast* allele for each gene was obtained independently.

In the Fig.5e we calculated allelic activity of a gene as a relation of *M.musculus* allele to *M.castaneus* allele. As the difference between *musculus* and *castaneus* is presumably minimal, the relation is expected to equal 1. That was the case for the *Kit* gene, and its expression relative to *castaneus* was close to 1 in both Wt and *Kit* Δ 30k. But for the *Pdgfra* gene the relation turned out to be more than 1, and the data was more dispersed.

The *Kit* Δ 30k deletion caused *Kdr* activation. In the experiment only *musculus* allele carried the deletion, consequently in that case the *musculus/castaneus* relation in *Kit* Δ 30k was much higher than Wt. As the effect of gene activation greatly changed the representation of the *Kdr* transcript, we detected at least 20 times higher relation score.

We demonstrated the logic of the experiment at Fig.5d. In that example in case of Wt RNA-seq demonstrated equal numbers of *musculus* (black) and *castaneus* (orange) transcripts and *musculus/castaneus* relation equaled 1. But in case of *Kit* Δ 30k RNA-seq revealed that the number of *musculus* transcripts is higher than *castaneus* transcripts, consequently the relation equaled 2.

To clarify the possible confusion we updated an example at the Fig.5d to highlight the *musculus* to *castaneus* relation rather than dependence of a genotype, as it was in the previous version.

We also added dot-plots in the Fig.5e to better visualize the data. For the updated figure we calculated *musculus/castaneus* relation for each of SNPs (as opposed to all the SNPs for one gene combined in the previous version). The minor change is that for the *Pdgfra* gene Wt and *Kit* Δ 30k are not that different in the updated version (in the previous version the contribution of one replica massively influenced the combined data). But the main result was not changed due to the different comparison method.

We hope that the provided explanation and revision of the Fig.5d,e would resolve the confusion.

Reviewer #2 (Remarks to the Author):

The authors addressed my comments.

Reviewer #3 (Remarks to the Author):

I want to thank the authors for their thoughtful responses to the questions and concerns raised. The clarifications and introduced changes enhance the readability and potential significance of the manuscript.

Some suggestions for text editing:

Line 147-150.: Long sentence and the dependent clause, “, for a more proximal regulatory region was essential in melanocytes.”, is not well connected to the main sentence. I suggest splitting the sentence.

Thank you for the suggestion, we have revised the text accordingly:

“Two enhancer regions were identified upstream of Kit in mast cells and melanocytes (located at 147-154 kb and 21-28 kb, respectively). While distant regions were required for Kit expression in mast cells (upon their deletion Kit was not transcribed), a more proximal regulatory region was essential in melanocytes.”

Line 252-253: Replace “We did not reveal...” with e.g. “The analysis did not reveal...”

Thank you for the suggestion, we have revised the text accordingly.

Line 155 and line 375: Replace “... genomic data ...” with “... functional genomic data...” to put more emphasis on the type of data generated to capture the cell-type dynamics, i.e. cHiC, RNA-seq, ChIP-seq.

Thank you for the suggestion, we have revised the text accordingly.